# Adversarial perturbation based latent reconstruction for domain-agnostic self-supervised learning

## Abstract

Most self-supervised learning (SSL) methods rely on domain-specific pretext tasks and data augmentations to learn high-quality representations from unlabeled data. Development of those pretext tasks and data augmentations requires expert domain knowledge. In addition, it is not clear why solving certain pretext tasks leads to useful representations. Those two reasons hinder wider application of SSL to different domains. To overcome such limitations, we propose adversarial perturbation based latent reconstruction (APLR) for domain-agnostic self-supervised learning. In APLR, a neural network is trained to generate adversarial noise to perturb the unlabeled training sample so that domain-specific augmentations are not required. The pretext task in APLR is to reconstruct the latent representation of a clean sample from a perturbed sample. We show that representation learning via latent reconstruction is closely related to multi-dimensional Hirschfeld-Gebelein-Rényi (HGR) maximal correlation and has theoretical guarantees on the linear probe error. To demonstrate the effectiveness of APLR, the proposed method is applied to various domains such as tabular data, images, and audios. Empirical results indicate that APLR not only outperforms existing domain-agnostic SSL methods, but also closes the performance gap to domain-specific SSL methods. In many cases, APLR also outperforms training the full network in a supervised manner.

## 1 Introduction

Unsupervised deep learning has been highly successful in discovering useful representations in natural language processing (NLP) (Devlin et al., 2019; Brown et al., 2020) and computer vision (Chen et al., 2020; He et al., 2020). These methods define pretext tasks on unlabeled data so that unsupervised representation learning can be done in a self-supervised manner without explicit human annotations. The success of self-supervised learning (SSL) depends on domain-specific pretext tasks, as well as domain-specific data augmentations. However, the development of semantic-preserving data augmentations requires expert domain knowledge, and such knowledge may not be readily available for certain data types such as tabular data (Ucar et al., 2021). Furthermore, the theoretical understanding of why certain pretext tasks lead to useful representations remains fairly elusive Tian et al. (2021). Those two reasons hinder wider applications of SSL beyond the fields of NLP and computer vision.

Self-supervised algorithms benefit from inductive biases from domain-specific designs but they do not generalize across different domains. For example, masked language models like BERT (Devlin et al., 2019) are not directly applicable to untokenized data. Although contrastive learning does not require tokenized data, its success in computer vision cannot be easily leveraged in other domains due to its sensitivity to image-specific data augmentations (Chen et al., 2020). Furthermore, in contrastive learning, the quality of representations degrades significantly without those hand-crafted data augmentations (Grill et al., 2020). Inspired by denoising auto-encoding (Vincent et al., 2008; 2010; Pathak et al., 2016), perturbation of natural samples with Gaussian, Bernoulli, and mixup noises (Verma et al., 2021; Yoon et al., 2020) has been utilized as domain-agnostic data augmentations applicable for self-supervised representation learning of images, graphs, and tabular data. However, random noises may not be as effective since uniformly perturbing uninformative features may not lead to the intended goal of augmentations. Specifically, convex combinations in mixup noises (Zhang

et al., 2018; Yun et al., 2019) could generate out-of-distribution samples because there is no guarantee that the input data space is convex (Ucar et al., 2021). In this article, we use generative adversarial perturbation as a semantic-preserving data augmentation method (Baluja & Fischer, 2018; Poursaeed et al., 2018; Naseer et al., 2019; Nakkal & Salzmann, 2021) applicable to different domains of data. Adversarial perturbation is constrained by the $\ell_p$ norm distance to the natural sample so that it is semantic-preserving and does not change the label (Goodfellow et al., 2015; Madry et al., 2018).

With semantic-preserving perturbation, the pretext tasks in domain-agnostic SSL could be reconstruction of clean samples (Yoon et al., 2020) or instance discrimination of perturbed samples (Verma et al., 2021). Nevertheless, reconstruction of clean samples in the input space is computationally expensive because the input data dimension is high. Therefore, we present adversarial perturbation based latent reconstruction (APLR), a simple and intuitive domain-agnostic self-supervised pretext task derived from linear generative models, to learn representations from unlabeled data in a domain-agnostic manner. Contrary to the pretext task of instance discrimination, our method does not require comparison to a large number of negative samples to achieve good performance. The proposed APLR not only achieves strong empirical performance on SSL in various domains, but also has theoretical guarantees on the linear probe error on downstream tasks.

The contributions of this article are summarized below:

- We present adversarial perturbation based latent reconstruction for domain-agnostic self-supervised learning.
- The proposed APLR achieves strong linear probe performance on various data types without using domain-specific data augmentations.
- We provide theoretical guarantees on the linear probe error on downstream tasks.

## 2  BACKGROUND

**Learning representations** from two views of an input, $\mathbf{x}^1$ and $\mathbf{x}^2$, is appealing if the learned representations do not contain the noises in different views. This assumption can be explicitly encoded into the following **generative model** (Bach & Jordan, 2005) with one shared latent variable $\mathbf{z}$:

$$
\begin{aligned}
p(\mathbf{z}) &= \mathcal{N}\left(\mathbf{0}, \mathbf{I}\right) \\
p(\mathbf{x}^1 \mid \mathbf{z}) &= \mathcal{N}\left(\boldsymbol{\psi}^\top \mathbf{z}, \Sigma^1\right) \\
p(\mathbf{x}^2 \mid \mathbf{z}) &= \mathcal{N}\left(\boldsymbol{\eta}^\top \mathbf{z}, \Sigma^2\right),
\end{aligned}
\tag{1}
$$

where the model parameters $\boldsymbol{\psi}$, $\boldsymbol{\eta}$, $\Sigma^1$ and $\Sigma^2$ can be learned by maximum likelihood estimation. Reconstruction of input data via maximum likelihood estimation is computationally expensive when the dimension of the input data is high. Instead, the probabilistic generative model can be reinterpreted as **latent reconstruction**, which has the benefit of direct representation learning while retaining the properties of generative modeling.

To convert generative modelling into latent reconstruction, two assumptions need to be met. First, the assumption in generative modeling is that both datasets have similar low-rank approximations. In latent reconstruction, this can be achieved by correlating the pair of latent embeddings $\boldsymbol{\psi}\mathbf{x}^1$ and $\boldsymbol{\eta}\mathbf{x}^2$. Secondly, it is assumed in generative modelling that the latent variables follow an isotropic Gaussian distribution $p(\mathbf{z}) = \mathcal{N}\left(\mathbf{0}, \mathbf{I}\right)$. In latent reconstruction, the covariance matrix of the latent variables is diagonal, meaning that there is no covariance between different dimensions of the latent variable. This orthogonality constraint is equivalent to the assumption of isotropic Gaussian prior and avoids the trivial solution. As a result, by correlating the latent embeddings $\boldsymbol{\psi}\mathbf{x}^1$ and $\boldsymbol{\eta}\mathbf{x}^2$ and enforcing a diagonal covariance matrix, the properties of generative modelling can be retained in latent reconstruction.

The key principle behind latent reconstruction is that the latent representation of $\mathbf{x}^1$ is a good predictor for that of $\mathbf{x}^2$. Given two datasets $\mathbf{X}^1$ and $\mathbf{X}^2$ of $N$ observations, the projection directions are found by maximizing the regularized correlation between the latent scores of $\mathbf{x}^1$ and $\mathbf{x}^2$

$$
\max_{\boldsymbol{\psi}_i, \boldsymbol{\eta}_i} \frac{\mathrm{Cov}(\mathbf{X}^1 \boldsymbol{\psi}_i, \mathbf{X}^2 \boldsymbol{\eta}_i)^2}{\sqrt{\gamma + (1-\gamma)\mathrm{Var}(\mathbf{X}^1 \boldsymbol{\psi}_i)}\sqrt{\gamma + (1-\gamma)\mathrm{Var}(\mathbf{X}^2 \boldsymbol{\eta}_i)}},
\tag{2}
$$

where $\boldsymbol{\psi}_i$ and $\boldsymbol{\eta}_i$ are the $i$-th directions of the projection matrices and $0 \leq \gamma \leq 1$ is a regularization coefficient (Rosipal & Krämer, 2005; Hardoon et al., 2004). When $\gamma = 0$, it is unregularized canonical correlation analysis (CCA) (Hotelling, 1936). When $\gamma = 1$, it corresponds to partial least squares (PLS) (Wold, 1975; Wold et al., 1984). Solving the optimization problem in Eq. equation 2 requires singular value decomposition or non-linear iterative methods (NIPLAS algorithm) to make projection directions orthogonal to each other. The computation costs of both methods are prohibitively expensive when the data dimension is high or the number of samples is large. Therefore, it is more desirable to alternatively update $\boldsymbol{\psi}$ and $\boldsymbol{\eta}$ in an iterative manner (Breiman & Friedman, 1985).

# 3 ADVERSARIAL PERTURBATION BASED LATENT RECONSTRUCTION

## 3.1 LATENT RECONSTRUCTION

Let $\mathbf{x}^1$ be a perturbed sample with a certain type of noise, which is adversarial noise in our case. Our pretext task in SSL is to reconstruct the latent representation of the clean sample $\mathbf{x}^2$ from the perturbed sample $\mathbf{x}^1$. We use deep neural networks $\psi(\cdot)$ and $\eta(\cdot)$ to project $\mathbf{x}^1$ and $\mathbf{x}^2$ into latent spaces, respectively. The reconstruction in the latent space can be achieved by maximizing the inner product between $\psi\left(\mathbf{x}^1\right)$ and $\eta\left(\mathbf{x}^2\right)$, when $\psi\left(\mathbf{x}^1\right)$ and $\eta\left(\mathbf{x}^2\right)$ have zero mean and unit variance. Based on discussions in Section 2, latent reconstruction must be done with orthogonality constraints to avoid the trivial solution, in which $\psi(\cdot)$ and $\eta(\cdot)$ projects all input data into a constant vector. Latent reconstruction with orthogonality constraints is equivalent to finding the multi-dimensional Hirschfeld-Gebelein-Rényi (HGR) maximal correlation (Renyi, 1959; Makur et al., 2015) between two random views. It is defined as follows

$$\rho(\mathbf{x}^1; \mathbf{x}^2) \quad \triangleq \sup_{\substack{\mathbb{E}\left[\psi(\mathbf{x}^1)\psi(\mathbf{x}^1)^\top\right]=\mathbb{E}\left[\eta(\mathbf{x}^2)\eta(\mathbf{x}^2)^\top\right]=\mathbf{I} \\ \mathbb{E}[\psi(\mathbf{x}^1)]=\mathbb{E}[\eta(\mathbf{x}^2)]=\mathbf{0}}} \mathbb{E}\left[\psi\left(\mathbf{x}^1\right)^\top \eta\left(\mathbf{x}^2\right)\right], \tag{3}$$

where zero mean constraints can be easily satisfied using a batch normalization layer (Ioffe & Szegedy, 2015) and the constraints on identity covariance matrices can be achieved by forcing the off-diagonal elements to be zero. In practice, the constrained optimization problem in Eq. equation 3 is solved by minimizing the following loss

$$\mathcal{L}_{\text{LR}} = -\mathbb{E}_{\mathbf{x}^1,\mathbf{x}^2\in\mathcal{B}}\left[\psi\left(\mathbf{x}^1\right)^\top \eta\left(\mathbf{x}^2\right) + \frac{\gamma}{2}\left(\|\psi\left(\mathbf{x}^1\right)\psi\left(\mathbf{x}^1\right)^\top - \mathbf{I}\|_F^2 + \|\eta\left(\mathbf{x}^2\right)\eta\left(\mathbf{x}^2\right)^\top - \mathbf{I}\|_F^2\right)\right] \tag{4}$$

where $\gamma$ is a Lagrange multiplier, $\|\cdot\|_F$ denotes the Frobenius norm, and $\mathcal{B}$ is a mini-batch.

## 3.2 ADVERSARIAL PERTURBATION

Adversarial perturbation creates input samples that are almost indistinguishable from natural data but causes the deep learning models to make wrong predictions (Szegedy et al., 2014). We use a generative model to generate adversarial perturbation because it is capable of creating diverse adversarial perturbations very quickly (Baluja & Fischer, 2018; Poursaeed et al., 2018; Naseer et al., 2019; Li et al., 2020).

A generator $\mathcal{G}$ is trained to produce an unbounded adversarial $\mathcal{G}(\mathbf{x}^2) = \delta$. The perturbation is then clipped to be within an $\epsilon$ bound of $\mathbf{x}^2$ under the $\ell_p$ norm. Let $\mathbf{x}^1 = \mathbf{x}^2 + \delta$ be the perturbed view of the clean sample $\mathbf{x}^2$. The vast majority of adversarial perturbation methods rely on the classification boundary of the attacked neural network ($\psi(\cdot)$ and $\eta(\cdot)$) to train the generator via maximizing a cross-entropy loss. However, it is not possible to obtain the generative adversarial perturbation via maximizing a cross-entropy loss in our case because no label is available. In addition, existing generative adversarial perturbation methods explicitly relying on the classification boundary of the attacked model tend to over-fit to the training data (Nakkal & Salzmann, 2021). Instead of using a cross-entropy loss, we train $\mathcal{G}(\cdot)$ by maximizing the $\ell_2$ distance between $\psi(\mathbf{x}^1)$ and $\eta(\mathbf{x}^2)$

$$\mathcal{L}_{\text{adv}} = \mathbb{E}_{\mathbf{x}^1,\mathbf{x}^2\in\mathcal{B}} \|\eta(\mathbf{x}^2) - \psi(\mathbf{x}^1)\|^2, \tag{5}$$

where $\psi(\cdot)$ and $\eta(\cdot)$ are frozen.

### 3.3 Adversarial training

Our model is trained in an adversarial manner. Given a mini-batch of data, we train $\mathcal{G}(\cdot)$ by maximizing $\mathcal{L}_{\text{adv}}$ while freezing $\psi(\cdot)$ and $\eta(\cdot)$. Then we update the parameters in $\psi(\cdot)$ and $\eta(\cdot)$ alternatively by minimizing $\mathcal{L}_{\text{LR}}$ while freezing $\mathcal{G}(\cdot)$. This process is illustrated in Fig. 1 and Algorithm 1.

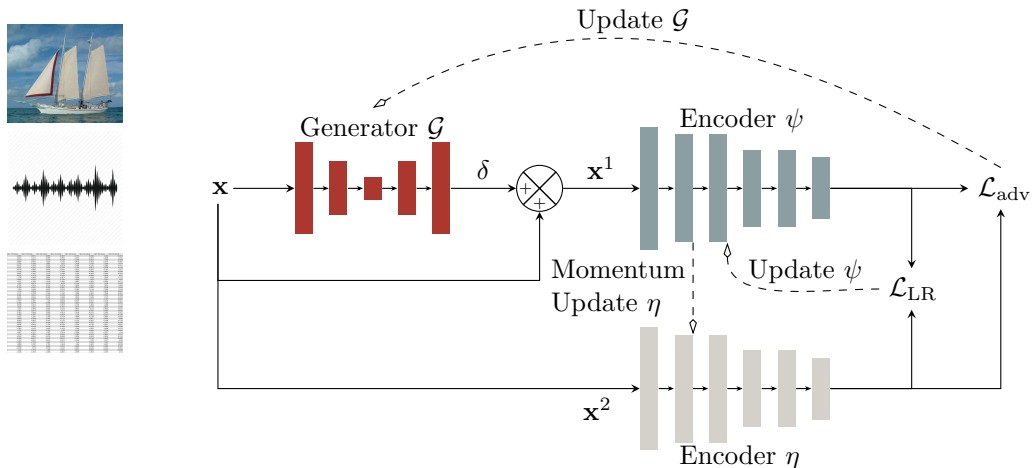

Figure 1: Illustrative diagram for adversarial perturbation based latent reconstruction. The adversarial generator network takes a clean sample as input and outputs $\delta$, an adversarial perturbation of the same shape. We constrain the perturbation by its $\ell_p$ norm and control its strength with the perturbation budget $\epsilon$. This constrained perturbation allows perturbed samples to appear unchanged to a human evaluator, making the perturbations semantic-preserving.

---

**Algorithm 1** Adversarial perturbation based latent reconstruction

Natural sample $\mathbf{x}$, encoders $\psi(\cdot)$ and $\eta(\cdot)$, noise generator $\mathcal{G}(\cdot)$, perturbation budget $\epsilon$, latent reconstruction loss $\mathcal{L}_{\text{LR}}$, adversarial loss $\mathcal{L}_{\text{adv}}$
**for** sampled minibatch **do**
    **Train** $\mathcal{G}(\cdot)$ (freeze $\psi(\cdot)$ and $\eta(\cdot)$)
        Generate an unbounded adversarial perturbation $\delta = \mathcal{G}(\mathbf{x})$     $\triangleright$ $\delta$ has the same shape as $\mathbf{x}$
        Clip adversarial perturbation $\delta = \epsilon\delta/|\delta|_p$
        Obtain the perturbed sample $\mathbf{x}^1 = \mathbf{x} + \delta$ and the clean sample $\mathbf{x}^2 = \mathbf{x}$
        Obtain latent representations $\psi(\mathbf{x}^1)$ and $\eta(\mathbf{x}^2)$
        Compute $\mathcal{L}_{\text{adv}}$ and update $\mathcal{G}(\cdot)$ using SGD
    **Train** $\psi(\cdot)$ **and** $\eta(\cdot)$ (freeze $\mathcal{G}(\cdot)$)
        Generate an unbounded adversarial perturbation $\delta = \mathcal{G}(\mathbf{x})$     $\triangleright$ $\delta$ has the same shape as $\mathbf{x}$
        Clip adversarial perturbation $\delta = \epsilon\delta/|\delta|_p$
        Obtain the perturbed sample $\mathbf{x}^1 = \mathbf{x} + \delta$ and the clean sample $\mathbf{x}^2 = \mathbf{x}$
        Obtain latent representations $\psi(\mathbf{x}^1)$ and $\eta(\mathbf{x}^2)$
        Compute $\mathcal{L}_{\text{LR}}$ and update $\psi(\cdot)$ using SGD
        Update $\eta(\cdot)$ using the exponentially moving average of parameters in $\psi(\cdot)$
**end for**

---

## 4 Theoretical Analysis

Let $\mathbf{x}$ be a data sample without perturbation and $y(\mathbf{x})$ be its downstream task label. The quality of the representation $\psi(\mathbf{x})$ is evaluated by the linear probe error, which is the linear classification

error of predicting $y(\mathbf{x})$ from $\mathbf{z}$ using a linear model parameterized by $\mathbf{B} \in \mathbb{R}^{k \times r}$. Let $f_B(\mathbf{x}) = \arg\max_{i \in [r]} (\psi(\mathbf{x})^\top \mathbf{B})_i$ be the prediction of the linear model. The linear probe error on $\psi(\mathbf{x})$ is defined as

$$\mathrm{Err}_\psi := \min_{\mathbf{B}} \Pr_{\mathbf{x} \sim P(\mathbf{x})} [y(\mathbf{x}) \neq f_B(\mathbf{x})], \tag{6}$$

where $P(\mathbf{x})$ is the data distribution.

We have to make two assumptions to bound the linear probe error on the learned representations. First, we assume that some universal minimizer of Eq. equation 3 can be realized by $\psi(\cdot)$ and $\eta(\cdot)$. When the nonlinear mapping to find multi-dimensional HGR maximal correlation is realizable by neural networks, we can analyze the quality of the learned representations using the properties in estimating the HGR maximal correlation.

**Assumption 4.1** (Realizability). *Let $\mathcal{H}$ be a hypothesis class containing functions $\psi : \mathcal{X}^1 \to \mathbb{R}^k$ and $\eta : \mathcal{X}^2 \to \mathbb{R}^k$. We assume that at least one of the global minima of $\mathcal{L}(\psi, \eta)$ belongs to $\mathcal{H}$.*

In addition, it is also reasonable to assume that an optimal classifier $f^*(\cdot)$ can predict the label of $\mathbf{x}$ almost deterministically under semantic-preserving perturbation. The assumption about the classification error of $f^*(\cdot)$ provides a baseline to quantify the linear probe error because part of the error is from approximating $f^*(\cdot)$ using a linear model.

**Assumption 4.2** ($\alpha$-bounded Error of the Optimal Classifier). *Let $\mathbf{x}$ be a data sample without perturbation and $y(\mathbf{x})$ be its downstream task label. $\delta$ is semantic-preserving perturbation. Then, we assume that there is a classifier $f^*$ such that $Pr(f^*(\mathbf{x}) \neq y(\mathbf{x})) \leq \alpha$ and $Pr(f^*(\mathbf{x} + \delta) \neq y(\mathbf{x})) \leq \alpha$.*

Given assumptions 4.1 and 4.2, we provide the following main theorem on the generalization bound when learning a linear classifier with finite labeled samples on the representations learned by maximizing the HGR maximal correlation.

**Theorem 4.3.** *Let $\psi^*, \eta^* \in \mathcal{H}$ be a minimizer of $\mathcal{L}(\psi, \eta)$. The linear classification parameter $\hat{\mathbf{B}}$ is estimated with $n_2$ i.i.d. random samples $\{(\mathbf{x}_i, y_i)\}_{i=1}^{n_2}$. With probability at least $1 - \zeta$ over the randomness of data, we have*

$$\Pr_{\mathbf{x} \sim P(\mathbf{x})} [y(\mathbf{x}) \neq f_{\hat{B}}(\mathbf{x})] \leq \widetilde{O}\left( \frac{\alpha}{1 - \lambda_{k+1}} + \sqrt{\frac{k}{n_2}} \right), \tag{7}$$

*where $\lambda_{k+1}$ is the $k + 1$-th Hirschfeld-Gebelein-Rényi maximal correlation between $\mathcal{X}^1$ and $\mathcal{X}^2$.*

We hide universal constant factors and logarithmic factors of $k$ in $\widetilde{O}(\cdot)$. The first term on the right hand side of Theorem 4.3 guarantees the existence of a linear classifier that achieves a small downstream classification error. It indicates whether the downstream label is linearly separable by the learned representation, thus measuring the expressivity of the learned representation. The second term on the right hand side reveals the sample complexity of learning $\mathbf{B}$ from finite labeled samples in the downstream task. It measures the data-efficiency of learning the downstream task using the learned representation. The proof is presented in the Appendix.

## 5 EXPERIMENTS

We demonstrate the effectiveness of the proposed method by application on three different data domains: tabular data, images, and audios. Additionally, we include ablation studies and sensitivity analysis in Appendix C. For all datasets, we follow the widely used linear evaluation protocol in self-supervised learning as a proxy to examine the quality of the learned representations (He et al., 2020; Chen & He, 2021). After the feature extractor is pretrained with unlabeled training data, we discard the projection-head, and learn a linear classifier on top of the frozen backbone network using the labeled training data.

For tabular and audio experiments, we search the perturbation budget hyperparameter $\epsilon$ from the set $\{0.05, 0.1, 0.15\}$. For image experiments, we fix $\epsilon$ to 0.05 for a direct comparison with Viewmaker networks (Tamkin et al., 2021). We find that constraining the perturbations to an $\ell_1$ norm distance

achieves the best results. For all experiments, we train the feature extractor and the adversarial generator in an alternating fashion. The feature extractor $\psi(\cdot)$ is trained with the SGD optimizer with momentum of 0.9 and weight decay of 5e-4. The learning rate is 0.03 without decay. The momentum coefficient in exponential moving average is set to 0.99 when updating $\eta(\cdot)$. The generator is trained with the Adam optimizer with an initial learning rate of 1e-3 and its architecture is described in the Appendix. Both the feature extractor and the generator are trained for 200 epochs with a batch size of 256. After self-supervised training on unlabeled data, a linear classifier is trained using SGD with a batch size of 256 and no weight decay for 200 epochs. The learning rate starts at 30.0 and is decayed to 0 after 200 epochs with a cosine schedule.

## 5.1 TABULAR DATA

For tabular data, we follow existing works (Yoon et al., 2020; Verma et al., 2021) and use MNIST and Fashion-MNIST as proxy datasets by flattening the images into 1-dimensional vectors. In addition, we use two real tabular datasets from the UCI repository to evaluate the proposed method (Dua & Graff, 2017).

**MNIST/Fashion-MNIST** are two image datasets of handwritten digits and Zalando's article images, respectively (LeCun et al., 2010; Xiao et al., 2017). The images of size $28 \times 28$ are flattened into vectors with 784 features. Both datasets have 10 classes, and contain 60,000 training examples and 10,000 test examples.

**Gas Concentrations** is a dataset containing chemical sensor measurements of 128 features when exposed to 6 different gases (Vergara et al., 2012; Rodriguez-Lujan et al., 2014). The classification task is to identify the target gas. We perform a 80/20 train/test split to obtain 11,128 training examples and 2,782 test examples.

**Gesture Phase** is a dataset containing 32 features extracted from videos of people in 5 different gestures (Wagner et al., 2014; Dua & Graff, 2017). We perform a 80/20 train/test split to obtain 7,898 training examples and 1,975 test examples.

For all tabular data experiments, we adopt a 10-layer MLP with residual connections Gorishniy et al. (2021) as the feature extractor. The generator is adapted from Nakkal & Salzmann (2021) by replacing convolutional layers with linear layers. The linear evaluation results on test datasets are reported in Table 1. APLR outperforms VIME-Self (Yoon et al., 2020), which corrupts tabular data and uses mask vector estimation and feature vector estimation as pretext tasks, on three out of four datasets. It aligns with the empirical observations that reconstruction of high-dimensional data in the input space is not necessary for learning high-quality representations. APLR outperforms the domain-agnostic benchmark DACL (Verma et al., 2021), which uses mixup noise, on all four datasets. Mixup noise is less effective than adversarial noise because mixup noise perturbs informative and uninformative dimensions in the input space uniformly. Furthermore, convex combinations in the input space via mixup may result in augmented views off the data manifold. Interestingly, our proposed APLR also outperforms training the full architecture in a supervised manner on the two real tabular datasets, Gas and Gesture.

Table 1: Linear evaluation accuracy on tabular data

|  | MNIST | Fashion-MNIST | Gas | Gesture |
|---|---|---|---|---|
| Tabular-specific |  |  |  |  |
| VIME-Self[1] (Yoon et al., 2020) | 96.62 | **87.26** | 93.17 | 38.99 |
| Domain-agnostic |  |  |  |  |
| DACL (Verma et al., 2021) | 94.70 | 79.78 | 95.39 | 38.56 |
| Ours | **97.11** | 87.12 | **97.98** | **42.97** |
| Fully supervised training | 98.67 | 90.00 | 94.91 | 42.32 |

---

[1]For experiments with VIME-Self, we align the experimental setup with APLR to use the same backbone architecture and perform pretraining on the entire training set.

## 5.2 IMAGE DATA

We use four benchmark image datasets to evaluate the effectiveness of the proposed method, including CIFAR-10/100 (Krizhevsky, 2009), STL-10 (Coates et al., 2011) and Tiny-ImageNet (Le & Yang, 2015). The details of the datasets are described in the Appendix.

ResNet18 (He et al., 2016) is adopted as the backbone network in the feature extractor. We adopt the generator in Nakkal & Salzmann (2021) for image data. We present the results for self-supervised representation learning on image data in Table 2. It is observed that APLR outperforms DACL (Verma et al., 2021) by a large margin, indicating that adversarial noise is a more effective semantic-preserving perturbation than mix-up noise in DACL. Interpolation of input samples via mix-up could lead to out-of-distribution training samples because the input data space may not be convex. Our method also achieves better performance than Viewmaker (Tamkin et al., 2021), which is a domain-agnostic self-supervised learning method by discriminating adversarially perturbed data. The adversarial noise in APLR is more robust because the training process of adversarial noise in APLR does not rely on the classification boundary between augmented samples (Nakkal & Salzmann, 2021). Furthermore, we also compare APLR against methods that use image augmentations (e.g. cropping, rotation, horizontal flipping), such as SimCLR (Chen et al., 2020). It is found in previous studies (Grill et al., 2020; Chen et al., 2020) that random crop is a crucial data augmentation towards learning high-quality representations for image data. However, it is impossible to create cropped views of images using adversarial perturbation because the adversarial noise is additive to the natural sample. Given the importance of random crop and the inability to create cropped views with adversarial perturbations, achieving comparable accuracies between APLR and SimCLR indicates that adversarial noise is a highly effective data augmentation method.

Table 2: Linear evaluation accuracy on image data

|  | CIFAR-10 | CIFAR-100 | STL-10 | Tiny-ImageNet |
|---|---|---|---|---|
| Image-specific |  |  |  |  |
| SimCLR (Chen et al., 2020) | **86.47** | 54.86 | 85.49 | **43.27** |
| Domain-agnostic |  |  |  |  |
| DACL (Verma et al., 2021)[2] | 60.49 | 35.28 | 57.34 | 22.69 |
| Viewmaker (Tamkin et al., 2021) | 84.51 | 52.28 | 82.73 | 40.51 |
| Ours | 85.92 | **55.83** | **86.21** | 42.93 |

## 5.3 AUDIO DATA

We use three benchmark audio datasets to evaluate APLR and describe the dataset details below.

**ESC-10/50** are two environmental sound classification datasets containing 5-seconds of environmental recordings (Piczak, 2015). ESC-10 and ESC-50 have 10 and 50 classes, and contain 400 and 2000 examples, respectively. We use the original fold settings from the authors (Piczak, 2015), and follow the experimental setup in Al-Tahan & Mohsenzadeh (2021) to use the first fold for testing and the rest for training.

**LibriSpeech-100** is a corpus of read English speech (Panayotov et al., 2015). We use speaker identification as the downstream classification task. We follow the experimental setup from Tamkin et al. (2021) to pretrain with the LibriSpeech-100 hour corpus which contains 28,539 examples, and perform linear evaluation on the LibriSpeech development set which contains 2,703 examples.

For audio experiments, we use 1-D ResNet18 (He et al., 2016) as the feature extractor and adopt the generator in Nakkal & Salzmann (2021) with one input channel. The time-frequency representation is a 2D log mel spectrogram, normalized to zero mean and variance. We report the results on audio data benchmarks in Table 3, and visualize examples from LibriSpeech-100 in the Appendix. APLR performs significantly better than CLAR (Al-Tahan & Mohsenzadeh, 2021), which experimented extensively with combinations of audio-specific augmentations and uses fade in/out and time masking as their best-performing audio augmentations. Compared to image augmentations, the data augmen-

---

[2]DACL reports better results using larger backbones and more training epochs. We just report the reproduced results under our experiment settings.

tations for audio data are relatively underexplored. Our results demonstrate the advantage of learning audio augmentations over manually designed augmentations. Our proposed method also outperforms both domain-agnostic methods, DACL and Viewmaker. DACL performs close to APLR on the simple yet small ESC-10 and ESC-50 datasets. However, it is unable to learn effective representations on the larger and significantly more complex LibriSpeech-100 dataset. Even though both APLR and Viewmaker use adversarial noise, APLR outperforms Viewmaker by a large margin across all benchmark datasets. This indicates the effectiveness of learning augmentations by maximizing the discrepancy between latent representations.

In Table 3, we also report results on training the full architecture in a supervised manner. We find that linear classifiers trained on top of the representations learned by APLR outperforms the supervised model on ESC-10, and closes the gap to ESC-50 compared to other benchmarks, demonstrating the ability for APLR to learn useful latent representations. Current state-of-the-art supervised approaches report high accuracies (over 94%) on the ESC-50 dataset (Gong et al., 2021; Kumar & Ithapu, 2020). However, these methods perform pretraining using large datasets such as AudioSet and ImageNet, and use multiple audio-specific data augmentations. With the supervised training experiments, we do not perform pretraining with large datasets, and we use time masking and frequency masking as augmentations. Our goal is to simply compare APLR against training the same architecture in a supervised manner.

Table 3: Linear evaluation accuracy on audio data

|  | ESC-10 | ESC-50 | LibriSpeech-100 |
|---|---|---|---|
| Audio-specific | | | |
| CLAR (Al-Tahan & Mohsenzadeh, 2021) | 68.70 | 40.40 | 62.14 |
| Domain-agnostic | | | |
| DACL (Verma et al., 2021) | 77.75 | 48.50 | 37.30 |
| Viewmaker (Tamkin et al., 2021) | 70.00 | 35.75 | 88.30 |
| Ours | **81.25** | **57.75** | **96.29** |
| Fully supervised training | 76.25 | 59.14 | N/A[3] |

## 6 RELATED METHODS

**Unsupervised representation learning** can be categorized into two classes based on the type of the pretext task: generative or discriminative. Generative approaches learn to generate or reconstruct unlabeled data in the input space (Higgins et al., 2017; Donahue et al., 2017; Donahue & Simonyan, 2019). Reconstructing the masked portion of data is highly successful in discovering useful representations in natural language processing (Mikolov et al., 2013; Brown et al., 2020; Devlin et al., 2019). Before the success of masked language models, variants of denoising or masked autoencoders are developed for computer vision tasks (Vincent et al., 2008; 2010; Pathak et al., 2016) but the performance is worse than discriminative SSL methods. It was not until recently that masked image models are revived in unsupervised visual representation learning by discretizing image patches via tokenizers (Bao et al., 2022; Zhou et al., 2022). MAE (He et al., 2022) further simplifies masked image models by directly inpainting masked images without tokenizers or image-specific augmentations. Although masking is a simple data augmentation that can be flexibly applied to different domains of data, computationally expensive generation or reconstruction in the input space may not be necessary for representation learning. Our method is derived from a generative model and shares the idea of reconstructing the corrupted samples in masked autoencoding. Instead of reconstructing discrete tokens or raw inputs, our method reconstructs the continuous latent representation, which is related to discriminative SSL methods using data augmentations.

**Augmentation-based discriminative SSL methods** learn representation by *comparing* (including but not limited to contrastive learning) augmented views of unlabeled data in the latent space. This line of work involves a contrastive framework with variants of InfoNCE loss (Gutmann & Hyvärinen, 2010;

---

[3]Following the experimental setup in Tamkin et al. (2021), the training set in LibriSpeech has 251 classes and the testing set has 40 classes. Since we train the fully supervised model in an end-to-end manner without a linear evaluation phase, we cannot report a result for LibriSpeech-100.

Oord et al., 2018) to pull together representations of augmented views of the same training sample (positive sample) and disperse representations of augmented views from different training samples (negative samples) (Wu et al., 2018; Hénaff et al., 2020; Wang & Isola, 2020). Typically, contrastive learning methods require a large size of negative samples to learn high-quality representations from unlabeled data (Chen et al., 2020; He et al., 2020). Meanwhile, non-contrastive methods train neural networks to match the representations of augmented positive pairs without comparison to negative samples or cluster centers. However, non-contrastive methods suffer from trivial solutions where the model maps all inputs to the same constant vector, known as a collapsed representation. Various methods have been proposed to avoid a collapsed representation on an ad hoc basis, such as asymmetric network architecture (Grill et al., 2020), stop gradient (Chen & He, 2021), and feature decorrelation (Ermolov et al., 2021; Zbontar et al., 2021; Hua et al., 2021). Interestingly, our method also includes a feature decorrelation constraint, which is adapted from a generative model. Recently, adversarial perturbations are combined with image augmentations to create more challenging positive and negative samples in self-supervised learning (Ho & Nvasconcelos, 2020; Yang et al., 2022). APLR does not require domain-specific augmentations and can be applied to different domains of data.

**Learning augmentations** has been investigated in supervised learning to obtain data-dependent augmentation policies for better generalization (Cubuk et al., 2019; Hataya et al., 2020). In parallel, adversarial perturbation can be treated as a special form of learnable augmentations to enhance the robustness of models with adversarial training (Goodfellow et al., 2015; Madry et al., 2018). The domain-agnostic augmentations in our method are closely related to generative adversarial perturbation, where data augmentations are obtained through a forward pass of learnable generative models (Li et al., 2020; Baluja & Fischer, 2018; Poursaeed et al., 2018; Naseer et al., 2019). The vast majority of adversarial perturbation methods rely on the classification boundary of the attacked neural network to train the generator via maximizing a cross-entropy loss. Those ideas have been extended to SSL to get adversarial perturbation by maximizing the InfoNCE loss in SimCLR (Kim et al., 2020; Tamkin et al., 2021). However, existing generative adversarial perturbation methods rely explicitly on the classification boundary or the instance discrimination boundary of the attacked model and tend to make them over-fit to the source data (Nakkal & Salzmann, 2021). Instead of maximizing a cross-entropy loss, we maximize the $\ell_2$ distance between mid-level feature maps to obtain generative adversarial perturbations.

**Theoretical understanding of SSL** has been studied under the assumption that augmented views of the same raw sample are somewhat conditionally independent given the label or a hidden variable (Arora et al., 2019; Tosh et al., 2021a;b; Lee et al., 2021). However, those assumptions do not hold in practice because augmented views of a natural sample are usually highly correlated. Augmented views are unlikely to be independent given the hidden label. Recent studies in contrastive learning provide theoretical guarantees of the learned representation without the assumption of conditional independence (HaoChen et al., 2021; Wang et al., 2022). In parallel, Tian et al. (2021) investigates the training dynamics of non-contrastive SSL methods to show how feature collapse is avoided but lacks guarantees for solving downstream tasks. Note that our proposed method does not involve an explicit comparison between positive and negative samples. Our theoretical analysis relies on the divergence transition matrix without the assumption of conditional independence.

## 7 CONCLUSIONS

In this article, we introduce APLR, a domain-agnostic SSL method by reconstruction of adversarial perturbed samples in the latent space. The adversarial perturbation is created by a generative network, which is trained concurrently with the feature encoder in an adversarial manner. Our empirical results show that the proposed method is better than the existing domain-agnostic SSL methods and achieves comparable performance with SOTA domain-specific SSL methods. In many cases, APLR also outperforms training the same architecture in a fully supervised manner, demonstrating its strong ability to learn useful latent representations. In addition, the proposed latent reconstruction is linked to Hirschfeld-Gebelein-Rényi maximal correlation and thus has theoretical guarantees of downstream classification tasks. We believe that the proposed method can be applied to applications beyond classification, such as reinforcement learning.

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

## A  PROOF OF THE MAIN THEOREM

The HGR maximal correlation can be estimated from divergence transition matrix $\mathbf{A} \in \mathbb{R}^{|\mathcal{X}^1| \times |\mathcal{X}^2|}$, whose entries are defined by the joint and marginal distributions of $\mathbf{x}^1$ and $\mathbf{x}^2$ (Witsenhausen, 1975). Let $P_{x^1}$ and $P_{x^2}$ be the marginal distribution and $P_{x^1 x^2}$ be the joint distribution. $P_{x^1}(\mathbf{x}_i^1)$ can be viewed as the probability mass of $\mathbf{x}_i^1$ being randomly sampled from $\mathcal{X}^1$. Then, each entry of $\mathbf{A}$ is given by

$$\mathbf{A}_{ij} = \frac{P_{x^1 x^2}(\mathbf{x}_i^1, \mathbf{x}_j^2)}{\sqrt{P_{x^1}(\mathbf{x}_i^1) P_{x^2}(\mathbf{x}_j^2)}} \tag{8}$$

The solution to Eq. equation 3 is the sum of the top $k$ singular values of $\mathbf{A}$ with left singular vectors $\mathbf{Z}^1 \in \mathbb{R}^{N \times k}$ and right singular vectors $\mathbf{Z}^2 \in \mathbb{R}^{N \times k}$ defined as

$$
\begin{aligned}
\mathbf{z}_i^1 &= \sqrt{P_{x^1}(\mathbf{x}_i^1)}\psi(\mathbf{x}_i^1), \quad i = 1, ..., N \\
\mathbf{z}_i^2 &= \sqrt{P_{x^2}(\mathbf{x}_i^2)}\eta(\mathbf{x}_i^2), \quad i = 1, ..., N
\end{aligned}
\tag{9}
$$

where $\mathbf{z}_i^1$ and $\mathbf{z}_i^2$ are the $i$-th row of embedding matrices $\mathbf{Z}^1$ and $\mathbf{Z}^2$, respectively (Huang & Xu, 2020). This is essentially a rank-$k$ approximation of $\mathbf{A}$ via minimizing $\|\mathbf{A} - \mathbf{Z}^1\mathbf{Z}^{2\top}\|_F^2$. Note that $\mathbf{x}^2$ is a clean sample and $\mathbf{z}^2$ is the representation of a clean sample. We use clean samples in downstream tasks. We drop the superscription to avoid cluttered notation.

The first term on the right hand side of the main theorem (theorem 4.3) measures the approximation error of the optimal classifier $f^*$ by a linear classifier parameterized by $\mathbf{B}$. It amounts to the residual of the least squares problem $\|\mathbf{f}^* - \mathbf{ZB}\|^2$ in Fig. 2, where the representation matrix $\mathbf{Z} \in \mathbb{R}^{N \times K}$ contains the top-$k$ left singular vectors of $\mathbf{A}$ and $\mathbf{f}^* \in \{0, 1\}^N$ is the vector that contains the predicted labels of all the data by the optimal classifier $f^*$. The approximation error is bounded if $\mathbf{f}^*$ has limited projection into the residual subspace that is perpendicular to the column space of the representation.

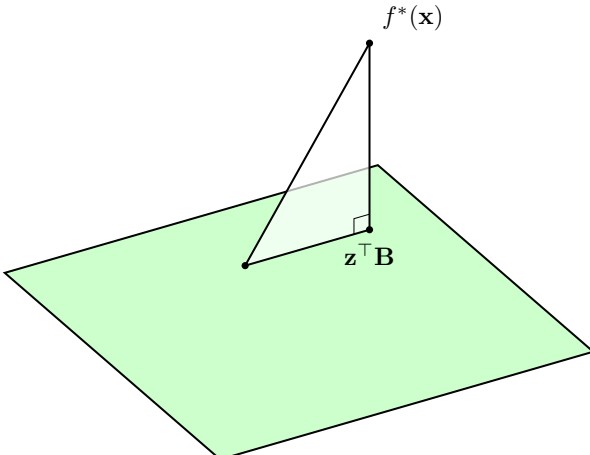

Figure 2: Geometric interpretation of least squares. $f^*(\mathbf{x}) : \mathcal{X} \rightarrow \{0, 1\}$ is the Bayes optimal classifier for predicting the label given $\mathbf{x}$ with error at most $\alpha$ according to Assumption 4.2. The green panel is the subspace spanned by the columns of the representation $\mathbf{z}$. $\mathbf{B}$ is the parameters of the linear classification model.

In the first step, we construct a quadratic form of $\mathbf{f}^*$ to quantify its projection into the residual space based on singular value decomposition of $\mathbf{A}$. The largest singular value of $\mathbf{A}$ is 1, with constant left and right singular vectors being $\mathbf{1}$ and $\mathbf{1}$ (Huang & Xu, 2020). Therefore, it is more convenient to subtract the top singular mode and introduce $\widetilde{\mathbf{A}} = \mathbf{I} - \mathbf{A}$. $\widetilde{\mathbf{A}}$ can be factorized as $\widetilde{\mathbf{A}} = \sum_{i=1}^{N} \gamma_i \mathbf{u}_i \mathbf{v}_i^\top$ via singular value decomposition, where $\gamma_i$ is the $i$-th singular value of $\widetilde{\mathbf{A}}$ with the left singular vector

$\mathbf{u}_i$ and the right singular vector $\mathbf{v}_i$. The quadratic form is given as follows

$$\mathbf{f}^{*\top}\widetilde{\mathbf{A}}\mathbf{f}^* = \mathbf{f}^{*\top}\left(\sum_{i=1}^{k}\gamma_i\mathbf{u}_i\mathbf{v}_i^\top + \sum_{i=k+1}^{N}\gamma_i\mathbf{u}_i\mathbf{v}_i^\top\right)\mathbf{f}^* \tag{10}$$

$$= \mathbf{f}^{*\top}\left(\sum_{i=1}^{k}\gamma_i\mathbf{u}_i\mathbf{u}_i^\top + \sum_{i=k+1}^{N}\gamma_i\mathbf{u}_i\mathbf{u}_i^\top\right)\mathbf{f}^* \tag{11}$$

$$\geq \mathbf{f}^{*\top}\left(\sum_{i=k+1}^{N}\gamma_i\mathbf{u}_i\mathbf{u}_i^\top\right)\mathbf{f}^* \tag{12}$$

$$\geq \mathbf{f}^{*\top}\left(\gamma_{k+1}\sum_{i=k+1}^{N}\mathbf{u}_i\mathbf{u}_i^\top\right)\mathbf{f}^* \tag{13}$$

$$= \gamma_{k+1}\mathbf{f}^{*\top}\mathbf{P}\mathbf{f}^* = \gamma_{k+1}\mathbf{f}^{*\top}\mathbf{P}^\top\mathbf{P}\mathbf{f}^* = \gamma_{k+1}\|\mathbf{P}\mathbf{f}^*\|^2 \tag{14}$$

where Eq. equation 11 is due to the fact that the left and right singular vectors are the same in the symmetric matrix $\widetilde{\mathbf{A}}$, the inequality in Eq. equation 12 is because of dropping a quadratic term, and Eq. equation 13 is due to $\gamma_{k+1} \leq \gamma_{k+2} \leq \ldots \gamma_N$. $\mathbf{P} \triangleq \sum_{i=k+1}^{N}\mathbf{u}_i\mathbf{u}_i^\top$ defines a projection matrix that projects $\mathbf{f}^*$ into a residual subspace spanned by singular vectors $\mathbf{u}_{k+1}, \ldots, \mathbf{u}_N$. Eq. equation 14 is obtained because $\mathbf{P}$ is an idempotent matrix ($\mathbf{P}^2 = \mathbf{P}$) (Greene, 2003). In addition, $(\mathbf{I} - \mathbf{P})\mathbf{f}^*$ is in the subspace spanned by singular vectors $\mathbf{u}_1, \ldots, \mathbf{u}_k$, which is the column space of $\mathbf{Z}$. Based on the geometric interpretation of the least squares problem $\|\mathbf{f}^* - \mathbf{Z}\mathbf{B}\|^2$, there exists $\mathbf{B}$ that such that $(\mathbf{I} - \mathbf{P})\mathbf{f}^* = \mathbf{Z}\mathbf{B}$ is the projection of $\mathbf{f}^*$ onto the column space of $\mathbf{Z}$.

In the second step, we upper bound $\gamma_{k+1}\|\mathbf{P}\mathbf{f}^*\|^2$. Based on Eq. equation 14, we have $\gamma_{k+1}\|\mathbf{P}\mathbf{f}^*\|^2 \leq \mathbf{f}^{*\top}\widetilde{\mathbf{A}}\mathbf{f}^*$. It is more convenient to upper bound $\mathbf{f}^{*\top}\widetilde{\mathbf{A}}\mathbf{f}^*$

$$\mathbf{f}^{*\top}\widetilde{\mathbf{A}}\mathbf{f}^* = \mathbf{f}^{*\top}\mathbf{I}\mathbf{f}^* - \mathbf{f}^{*\top}\mathbf{A}\mathbf{f}^*$$

$$= \sum_{i}^{N}\mathbf{f}_{x_i}^*\mathbf{f}_{x_i}^* - \sum_{i,j=1}^{N}P_{x^1x^2}(\mathbf{x}_i,\mathbf{x}_j)\left(\frac{\mathbf{f}_{x_i}^*}{\sqrt{P_{x^1}(\mathbf{x}_i)}}\frac{\mathbf{f}_{x_j}^*}{\sqrt{P_{x^2}(\mathbf{x}_j)}}\right)$$

$$= \frac{1}{2}\left(\sum_{i}^{N}\mathbf{f}_{x_i}^*\mathbf{f}_{x_i}^* - 2\sum_{i,j=1}^{N}P_{x^1x^2}(\mathbf{x}_i,\mathbf{x}_j)\left(\frac{\mathbf{f}_{x_i}^*}{\sqrt{P_{x^1}(\mathbf{x}_i)}}\frac{\mathbf{f}_{x_j}^*}{\sqrt{P_{x^2}(\mathbf{x}_j)}}\right) + \sum_{i}^{N}\mathbf{f}_{x_j}^*\mathbf{f}_{x_j}^*\right)$$

$$= \frac{1}{2}\left(\sum_{i}^{N}P_{x^1}(\mathbf{x}_i)\left(\frac{\mathbf{f}_{x_i}^*}{\sqrt{P_{x^1}(\mathbf{x}_i)}}\right)^2 - 2\sum_{i,j=1}^{N}P_{x^1x^2}(\mathbf{x}_i,\mathbf{x}_j)\left(\frac{\mathbf{f}_{x_i}^*}{\sqrt{P_{x^1}(\mathbf{x}_i)}}\frac{\mathbf{f}_{x_j}^*}{\sqrt{P_{x^2}(\mathbf{x}_j)}}\right)\right.$$

$$\left. + \sum_{j}^{N}P_{x^2}(\mathbf{x}_i)\left(\frac{\mathbf{f}_{x_j}^*}{\sqrt{P_{x^2}(\mathbf{x}_j)}}\right)^2\right) \tag{15}$$

$$= \frac{1}{2}\sum_{i}^{N}\sum_{j}^{N}P_{x^1x^2}(\mathbf{x}_i,\mathbf{x}_j)\left(\frac{\mathbf{f}_{x_i}^*}{\sqrt{P_{x^1}(\mathbf{x}_i)}} - \frac{\mathbf{f}_{x_j}^*}{\sqrt{P_{x^2}(\mathbf{x}_j)}}\right)^2,$$

where $\mathbf{f}_x^* = f^*(\mathbf{x})$, $P_{x^1}(\mathbf{x}_i) = \sum_j^N P_{x^1x^2}(\mathbf{x}_i,\mathbf{x}_j) = 1/N$ and $P_{x^2}(\mathbf{x}_j) = \sum_i^N P_{x^1x^2}(\mathbf{x}_i,\mathbf{x}_j) = 1/N$. Note that we only sample a pair of samples $(\mathbf{x}_i,\mathbf{x}_j)$ where $\mathbf{x}_i$ is created from semantic-preserving perturbation of $\mathbf{x}_j$ to train the model. The probability mass $P_{x^1x^2}(\mathbf{x}_i,\mathbf{x}_j) > 0$ only if $(\mathbf{x}_i,\mathbf{x}_j)$ are generated from a shared latent variable. Let $(\mathbf{x},\mathbf{x}^+)$ be a positive pair to denote a pair of samples created from semantic-preserving perturbation. We can rewrite equation equation 15 as

$$\mathbf{f}^{*\top}\widetilde{\mathbf{A}}\mathbf{f}^* = \frac{N}{2}\mathbb{E}_{x,x^+}[(\mathbf{f}_x^* - \mathbf{f}_{x^+}^*)^2], \tag{16}$$

where $\mathbb{E}_{x,x^+}[(\mathbf{f}_x^* - \mathbf{f}_{x^+}^*)^2]$ quantifies the probability that the optimal classifier $f^*(\cdot)$ predicts different labels for $(\mathbf{x},\mathbf{x}^+)$. When $\mathbf{f}_x^* \neq \mathbf{f}_{x^+}^*$, there must be $f^*(\mathbf{x}) \neq y(\mathbf{x})$ or $f^*(\mathbf{x}^+) \neq y(\mathbf{x})$. With Assumption 4.2, we have $\Pr(f^*(\mathbf{x}) \neq f^*(\mathbf{x}^+)) = 2\alpha$. As such, the quadratic form in Eq. equation 16 can be upper bounded:

$$\mathbf{f}^{*\top}\widetilde{\mathbf{A}}\mathbf{f}^* \leq N\alpha. \tag{17}$$

With Eq. equation 14 and equation 17, we have

$$\|\mathbf{f}^* - \mathbf{Z}\mathbf{B}\|^2 = \|\mathbf{P}\mathbf{f}^*\|^2 \leq \frac{N\alpha}{\gamma_{k+1}} \tag{18}$$

$$\frac{1}{N}\|\mathbf{f}^* - \mathbf{Z}\mathbf{B}\|^2 \leq \alpha/\gamma_{k+1} \tag{19}$$

The $k$-th singular value of $\mathbf{A}$ is $\lambda_k = 1 - \gamma_k$, which is also the $k$-th Hirschfeld-Gebelein-Rényi maximal correlation by definition. Therefore, we have $\Pr_{\mathbf{x} \sim P(\mathbf{x})}[y(\mathbf{x}) \neq f_B(\mathbf{x})] \leq \widetilde{O}\left(\frac{\alpha}{1-\lambda_{k+1}}\right)$.

The second term on the right hand side of the main theorem is the estimation error that measures sample complexity of learning $\mathbf{B}$ with access to $n_2$ i.i.d. training samples in the downstream task. It can be upper bounded using the Rademacher complexity of linear models. Let $\mathcal{H}_1 = \{\mathbf{z} \rightarrow \mathbf{z}^\top \mathbf{B} : \|\mathbf{B}\|_F \leq C_b\}$. We have the Rademacher complexity of the linear model

$$R_{n_2}(\mathcal{H}_1) = \frac{C_b\sqrt{C_z}}{\sqrt{n_2}} \tag{20}$$

where $\mathbb{E}[\|\mathbf{z}\|^2] \leq C_z$. By definition of Eq. equation 3, $\mathbb{E}[\|\mathbf{z}\|^2]$ captures the summation of first $k$ HGR maximal correlation. $\mathbb{E}[\|\mathbf{z}\|^2] \leq k$ because the HGR maximal correlation less equal than 1. Therefore, we have

$$\Pr_{\mathbf{x} \sim P(\mathbf{x})}[y(\mathbf{x}) \neq f_{\hat{B}}(\mathbf{x})] \leq \widetilde{O}\left(\frac{\alpha}{1-\lambda_{k+1}} + \sqrt{\frac{k}{n_2}}\right).$$

## B  IMAGE DATASETS

**CIFAR-10/100** are two datasets of tiny natural images with a size of $32 \times 32$ (Krizhevsky, 2009). CIFAR-10 and CIFAR-100 have 10 and 100 classes, respectively. Both datasets contain 50,000 training images and 10,000 test images.

**STL-10** is a 10-class image recognition dataset for unsupervised learning (Coates et al., 2011). Each class contains 500 labeled training images and 800 test images. In addition, it also contains 100,000 unlabeled training images. Both labeled and unlabeled training images are used for feature extractor pretraining without using labels. The linear classifier is learned using the labeled training images.

**Tiny-ImageNet** is a subset of the ILSVRC-2012 classification dataset (Le & Yang, 2015). It consists of 200 classes, with 500 training images, 50 validation images, and 50 test images in each class. The size of each image is $64 \times 64$.

## C  ADDITIONAL RESULTS

To understand the effectiveness of adversarial perturbations within APLR, we perform several additional experiments. First, we compare perturbation by adversarial noise against perturbation by Gaussian noise and random masking. For image datasets, we additionally compare the proposed adversarial perturbations against common image augmentations used in supervised learning, including CutMix (Yun et al., 2019), RandAugment (Cubuk et al., 2019), and Random Erasing (Zhong et al., 2017). Next, we explore the sensitivity of APLR to different perturbation strengths and Lagrange multipliers. Lastly, we compare our framework against SOTA SSL methods on image data.

### C.1  ABLATION STUDY

First, for all datasets, we perform ablations to compare perturbations with adversarial noise against Gaussian noise and masking. To obtain a sample augmented with Gaussian noise, we use $\mathbf{x}^1 = \mathbf{x}^2 + \delta$, where $\delta \sim \mathcal{N}\left(\mathbf{0}, \sigma^2 \mathbf{I}\right)$. For each dataset, we search the standard deviation $\sigma$ from the set $\{1, 3, 5, 10\}$ and report the best linear evaluation accuracy. For experiments with masking, we randomly mask a proportion of the clean sample $\mathbf{x}^2$. We search the proportion of masking from the set $\{20\%, 40\%, 50\%, 60\%, 70\%\}$ and report the best linear evaluation accuracy.

Tables 4 - 6 summarize the results. The adversarial noise outperforms the Gaussian noise and random masking on all datasets, except MNIST. Random noises may not be as effective since uniformly perturbing uninformative features may not lead to the intended goal of augmentations. That is why APLR leads to significant improvement over random perturbations on complex data, such as images and audios.

Table 4: Ablation study on tabular data.

|  | MNIST | Fashion-MNIST | Gas | Gesture |
|---|---|---|---|---|
| Gaussian noise | 97.43 | 85.77 | 96.25 | 41.46 |
| Masking | **97.58** | 86.95 | 95.70 | 42.30 |
| APLR | 97.11 | **87.12** | **97.98** | **42.97** |

Table 5: Ablation study on image data.

|  | CIFAR-10 | CIFAR-100 | STL-10 | Tiny-ImageNet |
|---|---|---|---|---|
| Gaussian Noise | 53.58 | 28.43 | 52.76 | 12.07 |
| Masking | 48.79 | 27.43 | 50.39 | 11.29 |
| APLR | **85.92** | **55.83** | **86.21** | **42.93** |

Table 6: Ablation study on audio data.

|  | ESC-10 | ESC-50 | LibriSpeech-100 |
|---|---|---|---|
| Gaussian noise | 75.00 | 41.75 | 78.52 |
| Masking | 77.50 | 45.00 | 76.76 |
| APLR | **81.25** | **57.75** | **96.29** |

Additionally, we compare adversarial noise against image augmentations in supervised learning, namely CutMix (Yun et al., 2019), RandAugment (Cubuk et al., 2019), and Random Erasing (Zhong et al., 2017). The results are summarized in Table 7. Random Erasing results in the worst performance among all methods, while CutMix is on par with Mixup in SSL. This is expected because CutMix performs slightly better or similar to MixUp in supervised learning. RandAugment leads to better performance than CutMix and MixUp because RandAugment contains a wide range of image augmentations. However, RandAugment does not outperform SimCLR. The studies in SimCLR show that careful selection of image augmentations is necessary for good performance in SSL. Some effective image augmentations in supervised learning do not lead to good performance in SSL.

Table 7: Additional ablation study on image data.

|  | CIFAR-10 | CIFAR-100 | STL-10 | Tiny-ImageNet |
|---|---|---|---|---|
| SimCLR | **86.47** | 54.86 | 85.49 | **43.27** |
| CutMix | 61.27 | 35.14 | 58.33 | 22.83 |
| RandAugment | 84.34 | 51.92 | 84.37 | 40.68 |
| Random Erasing | 56.71 | 28.89 | 55.91 | 18.45 |
| Ours | 85.92 | **55.83** | **86.21** | 42.93 |

## C.2 SENSITIVITY ANALYSIS

We perform experiments to understand how sensitive APLR is to different strengths of the adversarial perturbation and Lagrange multiplier.

We experiment with perturbation strengths of $\epsilon \in \{0.05, 0.1, 0.15\}$, and report the results in Table 8. The sensitivity analysis indicates that our method is robust to the adversarial perturbation strengths.

For the Lagrange multiplier, we experiment with $\gamma \in \{0.1, 0.5, 0.1\}$ and report the results in Table 9. We find that our method is robust to $\gamma$ and achieves strong performance. For APLR, we selected $\gamma = 1$ as the default value since it performed well consistently.

Table 8: Sensitivity to adversarial perturbation strengths.

| | $\epsilon = 0.05$ | $\epsilon = 0.1$ | $\epsilon = 0.15$ |
|---|---|---|---|
| Tabular Data | | | |
| MNIST | 97.11 | 96.15 | 93.73 |
| Fashion-MNIST | 87.04 | 86.40 | 84.14 |
| Gas | 97.50 | 97.19 | 97.98 |
| Gesture | 42.97 | 40.90 | 41.46 |
| Image Data | | | |
| CIFAR-10 | 85.92 | 84.66 | 85.26 |
| CIFAR-100 | 55.83 | 54.37 | 54.77 |
| STL-10 | 86.21 | 85.04 | 85.64 |
| Tiny-ImageNet | 42.93 | 42.42 | 41.47 |
| Audio Data | | | |
| ESC-10 | 78.75 | 81.25 | 77.50 |
| ESC-50 | 54.25 | 54.50 | 57.75 |
| LibriSpeech-100 | 93.55 | 96.29 | 96.29 |

Table 9: Sensitivity to Lagrange multiplier.

| | $\gamma = 0.1$ | $\gamma = 0.5$ | $\gamma = 1$ |
|---|---|---|---|
| Tabular Data | | | |
| MNIST | 96.54 | 96.93 | 97.11 |
| Fashion-MNIST | 86.83 | 86.69 | 87.12 |
| Gas | 84.61 | 97.97 | 97.98 |
| Gesture | 40.35 | 40.35 | 42.97 |
| Audio Data | | | |
| ESC-10 | 80.00 | 75.00 | 81.25 |
| ESC-50 | 47.75 | 44.50 | 57.75 |
| LibriSpeech-100 | 89.45 | 87.30 | 96.29 |

Our sensitivity analyses indicate that our method is robust to hyperparameters such as $\epsilon$ and $\gamma$. The proposed APLR achieves strong performance as long as the hyperparameter values are within reasonable ranges.

## C.3 APLR AGAINST SOTA IMAGE-SPECIFIC SSL METHODS

We perform an analysis to compare the proposed framework against SOTA SSL methods on images, namely SimCLR (Chen et al., 2020), Barlow Twins (Zbontar et al., 2021), and BYOL (Grill et al., 2020). For this experiment, we use the image augmentations described in SimCLR (Chen et al., 2020) for a fair comparison against image-specific SSL methods. We train each model for 200 epochs and summarize the results in Table 10. Our method achieves comparable performance to BYOL and Barlow Twins.

Table 10: APLR vs. SOTA SSL methods on image data

| | CIFAR-10 | CIFAR-100 | STL-10 | Tiny-ImageNet |
|---|---|---|---|---|
| SimCLR | 86.47 | 54.86 | 85.49 | 43.27 |
| Barlow Twins | 89.02 | **62.84** | 85.43 | **45.33** |
| BYOL | 88.54 | 61.76 | 85.59 | 42.75 |
| Ours | **89.63** | 62.55 | **86.41** | 44.76 |

## D    VISUALIZATIONS OF ORIGINAL AND PERTURBED SPECTROGRAMS

In Figure 3, we visualize random spectrograms from LibriSpech-100 and the deltas between the original and perturbed spectrograms. The perturbations are indistinguishable and thus semantic-preserving.

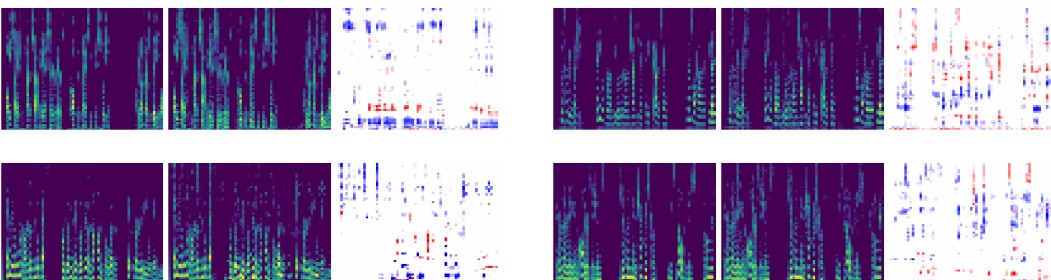

Figure 3: Examples triplets of original spectrograms (left), perturbed spectrograms (middle) and their differences (right) from LibriSpeech-100. The color scales for original and perturbed spectrograms are set to the scale of the original spectrogram. The color scale for the differences is set to -2.5 (red) to + 2.5 (blue), though some values exceed this range. Best viewed when zoomed.

## E    ADVERSARIAL GENERATOR ARCHITECTURE

The architecture of the generator is described in Table 11. For experiments on tabular data, we replace the convolution layers with fully connected layers.

Table 11: Architecture of the adversarial generator.

| Layer | Number of Filters | Kernel Size |
|---|---|---|
| Convolution Layer | 32 | 9 |
| Convolution Layer | 64 | 3 |
| Convolution Layer | 128 | 3 |
| Residual Block | 128 | 3 |
| Residual Block | 128 | 3 |
| Residual Block | 128 | 3 |
| Upsampling Convolution Layer (Upsample = 2) | 64 | 3 |
| Upsampling Convolution Layer (Upsample = 2) | 32 | 3 |
| Convolution Layer | | |

## F    VISUALIZATIONS OF TRAINING OBJECTIVES DURING TRAINING

To understand how well the final objective in Equation 4 approximates the HGR correlation, we plot the two loss terms over training in Figure 4. We find that the orthogonal loss approaches zero during training.

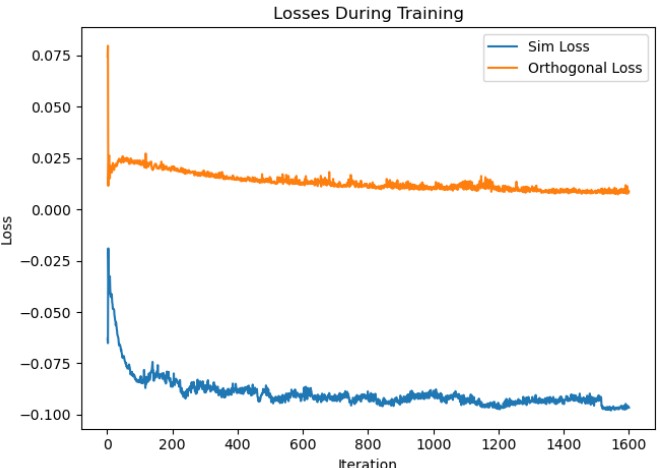

Figure 4: Losses during training.

