# OpenReview forum: "Adversarial perturbation based latent reconstruction for domain-agnostic self-supervised learning"
_ICLR.cc/2023/Conference — Submitted to ICLR 2023_

### Official Review · Reviewer_RgE2 · 2022-10-24

**Confidence:** 4
**Correctness:** 4
**Technical Novelty And Significance:** 2
**Empirical Novelty And Significance:** 2
**Recommendation:** 5

**Clarity, Quality, Novelty And Reproducibility:**

The paper is clear and easy to follow along. The idea proposed is slightly novel.

**Strength And Weaknesses:**

Strength -

1. The paper is generally easy to follow along and well-motivated.
2. The approach also show theoretical guarantees on the linear probe error for their approach.
3. Empirically the results are good across different domains and datasets.

Weakness -

1. I feel the paper's approach is very similar to BYOL, which also has a similar pretext task of reconstructing the latent representation of one augmented sample from another augmented image. The main difference is that BYOL uses domain augmentations while the proposed approach uses adversarial perturbation. However, there is no comparison with BYOL, at least for the image data. Further to avoid learning a trivial solution BYOL uses a predictor network on top of only one encoder(and hence asymmetric). In contrast, the paper injects an orthogonality constraint on the latent reconstructions.  How do these two approaches differ? Can authors do an ablation study between these two approaches? In particular, why would one want to use the orthogonal projection constraint as mentioned in the paper over the asymmetric networks approach as done in BYOL?

2. For learning the adversarial perturbation, the paper learn a generator to produce the perturbation. However, we could also instead just use the gradient of L_adv to compute the perturbation direction, similar to single-step PGD attack[1] done for adversarial attacks. This is much simpler than learning a generator. Can the authors comment on this, and show some ablation to validate that learning a generator is indeed better?

3. The proposed approach is still not domain agnostic as it cannot be applied over the text domain as it would be hard to define a perturbation over the input. This questions the domain-agnostic claim made in the paper. Masked learning approaches in contrast can be applied over both text and image data[2]. Can the authors comment on this?

References -
[1] Towards Deep Learning Models Resistant to Adversarial Attacks. Madry et al.
[2] Image as a Foreign Language: BEiT Pretraining for All Vision and Vision-Language Tasks. Wang et al.


**Summary Of The Paper:**

The paper proposes a domain-agnostic self-supervised learning algorithm using adversarial perturbations in the latent space. The paper argues that existing self-supervised learning algorithms either have domain-dependent pretext tasks or have domain dependent augmentations. For instance, the masked image/language models approach requires data to be tokenized while the contrastive learning approaches require domain-dependent augmentation. Thus, to propose a domain-agnostic approach, the authors devise a pre-text task where the objective is to reconstruct the latent representation of a clean sample from the perturbed sample. For perturbation, instead of using an augmentation the paper rather uses adversarial perturbation, where the perturbation is produced by a generator. The proposed algorithm iteratively switches between learning a generator to produce the adversarial perturbation and learning the encoder to predict the latent representation of the clean sample from the adversarially perturbed sample. To avoid learning the trivial solution by the encoder, latent reconstruction is done with an orthogonality constraint where the encoders project all input data into a constant vector. To show the generality of their approach, the authors present results over tabular data, audio data, and image data.




**Summary Of The Review:**

While I like the direction followed in the paper about proposing a domain-agnostic self-supervised learning approach I have concerns about the experimental setup, particularly proper comparisons with BYOL and also the method for doing adversarial perturbation. Furthermore, the proposed approach is still difficult to apply over text-domain and hence it is not domain-agnostic.

---

> ### Author Response · Authors · 2022-11-18
> **Response to Reviewer RgE2**
>
> We thank the reviewer for the insightful comments. Below is our point-to-point response.
>
> **Comment**: I feel the paper's approach is very similar to BYOL, which also has a similar pretext task of reconstructing the latent representation of one augmented sample from another augmented image. The main difference is that BYOL uses domain augmentations while the proposed approach uses adversarial perturbation. However, there is no comparison with BYOL, at least for the image data. Further to avoid learning a trivial solution BYOL uses a predictor network on top of only one encoder(and hence asymmetric). In contrast, the paper injects an orthogonality constraint on the latent reconstructions. How do these two approaches differ? Can authors do an ablation study between these two approaches? In particular, why would one want to use the orthogonal projection constraint as mentioned in the paper over the asymmetric networks approach as done in BYOL?
>
> **Answer**: The goal of our study is not to develop a new SSL method with some tricks to outperform the SOTA SSL methods. Instead, we want to build the bridge between current SSL methods to some well-understood statistical learning methods, such as CCA and PLS. We also want to show some insights on why minimizing certain loss functions using unlabeled data leads to useful representations in the downstream task. We believe such insights are very important for the development of unsupervised representation learning in the long term.
>
> We derive a deep learning based SSL method from CCA/PLS by replacing the linear projection with the nonlinear projection. The orthogonal constraint is a natural component of the loss function by converting the constrained optimization problem in CCA/PLS via Lagrange multiplier. The orthogonal constraint naturally prevents the collapsed representation. In addition, our formulation of SSL is linked to HGR maximal correlation. It is due to the link to HGR maximal correlation that we can derive some theoretical results about the proposed SSL method.
>
> In contrast, BYOL uses asymmetric networks to avoid collapsed representation. To the best of our knowledge, it is still elusive why asymmetric networks can avoid collapsed representation (Tian et al., 2021). Our method has good theoretical results to support why it works in SSL and how well it works for downstream tasks.
>
> We do the comparison with Barlow Twins and BYOL. We use the image augmentations described in SimCLR. All models are trained for 200 epochs. Our method achieves comparable performance to BYOL and Barlow Twins. Results have been added to Appendix C in the paper.
>
>
> |                 | CIFAR-10 | CIFAR-100 | STL-10 | Tiny-ImageNet |
> |-----------------|----------|-----------|--------|---------------|
> | SimCLR          | 86.47    | 54.86     | 85.49  | 43.27 |
> | Barlow Twins    | 89.02    | **62.84**    | 85.43 | **45.33**       |
> | BYOL       | 88.54   | 61.76     | 85.59  | 42.75        |
> | Our            | **89.63**    | 62.55    | **86.41**  | 44.76 |
>
> Tian, Yuandong, Xinlei Chen, and Surya Ganguli. "Understanding self-supervised learning dynamics without contrastive pairs." *International Conference on Machine Learning.* PMLR, 2021.
>
> **Comment**: For learning the adversarial perturbation, the paper learn a generator to produce the perturbation. However, we could also instead just use the gradient of L_adv to compute the perturbation direction, similar to single-step PGD attack[1] done for adversarial attacks. This is much simpler than learning a generator. Can the authors comment on this, and show some ablation to validate that learning a generator is indeed better?
>
> [1] Towards Deep Learning Models Resistant to Adversarial Attacks. Madry et al.
>
>
> **Answer**:  PGD attack involves direct compution of gradients with respect to the input data through the target backbone network, which can be computationally expensive especially when the target backbone network is large. With a generator, attacking the target network only involves a forward pass through the generator. Typically, the size of the generator is much smaller than that of the target network. In addition, generator based methods also achieve better empirical performance in adversarial attack (Baluja and Fischer, 2017, Poursaeed et al., 2018,  Nakkal and Salzmann, 2021).
>
> Baluja, Shumeet, and Ian Fischer. "Adversarial transformation networks: Learning to generate adversarial examples." arXiv preprint arXiv:1703.09387 (2017).
>
> Omid Poursaeed, Isay Katsman, Bicheng Gao, and Serge Belongie. Generative adversarial perturbations. In *Proceedings of the IEEE Conference on Computer Vision and Pattern Recognition*, pages 4422–4431, 2018.
>
> Krishna Kanth Nakkal and Mathieu Salzmann. Learning transferable adversarial perturbations. *Advances in Neural Information Processing Systems*, 34, 2021.

---

> > ### Author Response · Authors · 2022-11-18
> > **Response to Reviewer RgE2 (Continued)**
> >
> >
> > **Comment**: The proposed approach is still not domain agnostic as it cannot be applied over the text domain as it would be hard to define a perturbation over the input. This questions the domain-agnostic claim made in the paper. Masked learning approaches in contrast can be applied over both text and image data [2]. Can the authors comment on this?
> >
> > [2] Image as a Foreign Language: BEiT Pretraining for All Vision and Vision-Language Tasks. Wang et al.
> >
> >
> > **Answer**: Masked language modeling has been highly successful in NLP with the development of BERT and GPT. Adversarial noise can be added to any data as perturbation including word embeddings, and therefore APLR (our method) should work in text domain. However, we did not perform experiments on text data because we do not expect our method to outperform SOTA masked language modeling methods (such as the highly successful BERT and GPT). Additionally, such large language models have been pre-trained on the web to realize their performance, and we do not have the computational resources to perform large-scale pre-training. Our goal is not to outperform SOTA methods in NLP or computer visions, but to extend the success of SSL to some overlooked domains, such tabular and audio data.
> >
> > Masking can be applied to both images and language data. When pertaining with vision-language data, it leads to incredible performance. However, we do not always have paired data in two different domains to enjoy the benefit of multimodality pretraining. It is not directly comparable to single-domain unsupervised pretraining. Compared with SimCLR and BYOL, the original BEiT (Bao et al, 2022) achieves very weak linear probe performance on image data (Table 9 in their Appendix). In our study, we use linear probe to evaluate the downstream task performance. Currently, it is still debatable whether linear probe or end-to-end fine-tuning is the best way to measure the quality of the pretraining. We prefer linear probe because it only require training a linear model and can achieve strong performance when the number of training samples is small in down-stream tasks. We do not choose the masked learning due to weak linear probe performance.
> >
> > Bao, Hangbo, et al. "BEiT: BERT Pre-Training of Image Transformers." *International Conference on Learning Representations*. 2022.

---

### Official Review · Reviewer_Hv4e · 2022-10-24

**Confidence:** 4
**Correctness:** 3
**Technical Novelty And Significance:** 3
**Empirical Novelty And Significance:** 3
**Recommendation:** 8

**Clarity, Quality, Novelty And Reproducibility:**

Clarity and Quality
- Clarity and quality are adequate. There is no description of ζ around Eq. (7). Is there any relationship between the given bounds and ζ?

Novelty
- Modules to generate perturbed examples and train encoders under CCA/PLS setting are not novel. Their combination of self-supervised learning and theoretical discussion would be new.

Reproducibility
- How is the hyperparameter γ tuned? It appears to be an important parameter that controls whether it is CCA-like or PLS-like, but there seem to be no sensitivity experiments.
- The experimental setting is unclear. For example, the network architecture is described by citing Nakkal & Salzmann (2021), but the authors should try to keep the description self-contained for reproducibility.

**Strength And Weaknesses:**

Strengths
1. The proposed method is a simple but straightforward design based on probabilistic CCA and PLS and performs better than related methods.
1. Experiments have been conducted on data sets from multiple modalities, which is sufficient to demonstrate versatility.
1. Rather than ending with a heuristic proposal of a method, there is also a theoretical discussion.

Weaknesses
1. The abstract and introduction also mention data augmentation, but the experiments are limited to comparisons among self-supervised learning methods. For example, in the case of images, Cutmix, RandAugment, and Random Erasing are used in addition to Mixup, but no comparison with such data expansion methods is made. The reviewer agrees that there is no de-facto standard data augmentation for tabular data, but some comparisons using image and audio datasets would be possible.
1. Approaches that maximize correlation, such as CCA and PLS, are also reasonable, while recent trends, such as (Verma et al., 2021) and (Ho and Nvasconcelos, 2020), use contrast learning. A contrastive loss with original features x, x+δ, and features x' with different semantics seems natural. The reviewer would like to know if there is any motivation for the proposed method not to employ contrastive learning.

**Summary Of The Paper:**

This paper proposes a method for self-supervised learning based on adversarial sample generation. The proposed method is based on the probabilistic interpretation of PLS and/or CCA based on graphical models and does not require class labels or classifiers. It performs best among the domain-agnostic methods on tabular, image, and audio datasets. They also provide theoretical discussions as well as experimental comparisons.

**Summary Of The Review:**

Overall, the reviewers are inclined to judge this paper as acceptable. The score will be improved more if the above weaknesses and questions are adequately answered.

----

**Updated review**

Responses from the authors and additional experiments were sufficient to answer the reviewers' questions, thus improving the score. If there is room in the paper, the following responses from the authors could be added, even in summary.

> The fundamental motivation behind contrastive learning is maximizing the mutual information between two augmented views (Oord et al., 2018). However, it is inconsistent with the actual behavior in practice, e.g., optimizing a tighter bound on mutual information can lead to worse representations (Tschannen et al., 2019). Theoretical understanding of contrastive loss is still elusive. Analysis based on the assumption of latent classes provides nice theoretical insights (Saunshi et al., 2019), but the theoretical result that representation quality suffers with more negative samples is inconsistent with empirical results in SimCLR and popular contrastive learning methods. A recent study interprets contrastive loss as alignment and uniformity on the hypersphere (Wang et al, 2020), but they do not provide any theoretical guarantees on the downstream performance of the learned representation.

> For the reasons presented above, we decide not to employ contrastive learning and develop a self-supervised learning method based on well-known statistical learning methods such as CCA and PLS. Our goal is not to beat SOTA performance. We want to give some insights into why training models with unlabeled data using certain loss functions leads to useful representations in the downstream task. We believe such insights are very important for the development of unsupervised representation learning in the long term.

---

> ### Author Response · Authors · 2022-11-18
> **Response to Reviewer Hv4e**
>
> We thank the reviewer for the insightful comments. Below is our point-to-point response.
>
>
>
> **Comment**: The abstract and introduction also mention data augmentation, but the experiments are limited to comparisons among self-supervised learning methods. For example, in the case of images, Cutmix, RandAugment, and Random Erasing are used in addition to Mixup, but no comparison with such data expansion methods is made. The reviewer agrees that there is no de-facto standard data augmentation for tabular data, but some comparisons using image and audio datasets would be possible.
>
> **Answer**: In response to your comment, we did a quick comparison with CutMix, RandAugment, and Random Erasing, whose results are listed in the table below and can be found in Appendix C of the paper. Random Erasing results in the worst performance among all methods, while CutMix is on par with Mixup in SSL. This is expected because CutMix performs slightly better or similar to MixUp in supervised learning. RandAugment leads to better performance than CutMix and MixUp because RandAugment contains a wide range of image augmentations. However, RandAugment does not outperform SimCLR. The studies in SimCLR show that careful selection of image augmentations is necessary for good performance in SSL. Some effective image augmentations in supervised learning do not lead to good performance in SSL.
>
> |                | CIFAR-10 | CIFAR-100 | STL-10 | Tiny-ImageNet |
> |----------------|----------|-----------|--------|---------------|
> | SimCLR         | **86.47**    | 54.86     | 85.49  | **43.27**         |
> | CutMix         | 61.27    | 35.14     | 58.33  | 22.83         |
> | RandAugment    | 84.34    | 51.92     | 84.37  | 40.68         |
> | Random Erasing | 56.71    | 28.89     | 55.91  | 18.45         |
> | Ours           | 85.92    | **55.83** | **86.21**  | 42.93         |
>
> **Comment**: Approaches that maximize correlation, such as CCA and PLS, are also reasonable, while recent trends, such as (Verma et al., 2021) and (Ho and Nvasconcelos, 2020), use contrast learning. A contrastive loss with original features x, x+δ, and features x' with different semantics seems natural. The reviewer would like to know if there is any motivation for the proposed method not to employ contrastive learning.
>
> **Answer**: The fundamental motivation behind contrastive learning is maximizing the mutual information between two augmented views (Oord et al., 2018).  However, it is inconsistent with the actual behavior in practice, e.g., optimizing a tighter bound on mutual information can lead to worse representations (Tschannen et al., 2019). Theoretical understanding of contrastive loss is still elusive. Analysis based on the assumption of latent classes provides nice theoretical insights (Saunshi et al., 2019), but the theoretical result that representation quality suffers with more negative samples is inconsistent with empirical results in SimCLR and popular contrastive learning methods. A recent study interprets contrastive loss as alignment and uniformity on the hypersphere (Wang et al, 2020), but they do not provide any theoretical guarantees on the downstream performance of the learned representation.
>
> For the reasons presented above, we decide not to employ contrastive learning and develop a self-supervised learning method based on well-known statistical learning methods such as CCA and PLS. Our goal is not to beat SOTA performance. We want to give some insights into why training models with unlabeled data using certain loss functions leads to useful representations in the downstream task. We believe such insights are very important for the development of unsupervised representation learning in the long term.
>
> Oord, Aaron van den, Yazhe Li, and Oriol Vinyals. "Representation learning with contrastive predictive coding." arXiv preprint arXiv:1807.03748 (2018).
>
> Tschannen, Michael, et al. "On Mutual Information Maximization for Representation Learning." *International Conference on Learning Representations.* 2020.
>
> Saunshi, Nikunj, et al. "A theoretical analysis of contrastive unsupervised representation learning." *International Conference on Machine Learning*. PMLR, 2019.
>
> Wang, Tongzhou, and Phillip Isola. "Understanding contrastive representation learning through alignment and uniformity on the hypersphere." *International Conference on Machine Learning*. PMLR, 2020.

---

> > ### Author Response · Authors · 2022-11-18
> > **Response to Reviewer Hv4e (Continued)**
> >
> >
> > **Comment**: There is no description of ζ around Eq. (7). Is there any relationship between the given bounds and ζ?
> >
> > **Answer**: In Probably Approximately Correct (PAC) learning, ζ is called the confidence parameter or the failure probability. The error is bounded with high probability 1- ζ due to the randomness in the downstream training data. In linear probe, a linear classifier is learned. Based on the PAC learning theory on the linear classifier, it requires the number of training samples n2 in the downstream task greater than a threshold, which is the sample complexity of learning a model. The sample complexity depends on ln(1/ ζ). It means that the failure probability can be taken as a small number without hurting much on the error bound. It is omitted in the paper for brevity because it is the basics of the PAC learning theory for linear models.
> >
> >
> >
> >
> > **Comment**: How is the hyperparameter γ tuned? It appears to be an important parameter that controls whether it is CCA-like or PLS-like, but there seem to be no sensitivity experiments.
> >
> > **Answer**: We add an ablation study to understand how sensitive APLR is to different strengths of γ. We find that our method is robust to γ and achieves strong performance as long as the hyperparameter values are within reasonable ranges. For APLR, we selected γ = 1 as the default value since it performed consistently well.
> >
> >
> > |      $\gamma$  | MNIST        | Fashion-MNIST  | Gas     | Gesture        |
> > |:---------------|:---------------:|:---------------:|:---------------:|:---------------:|
> > | 0.1 | 96.54      | 86.83          | 84.61         | 40.35           |
> > | 0.5      | 96.93       | 86.69      | 97.97   | 40.35         |
> > | 1       | 97.11        | 87.12 | 97.98 | 42.97 |
> >
> >
> > |     $\gamma$               | ESC-10       | ESC-50         | LibriSpeech-100 |
> > |:----------------|:--------------:|:----------------:|:-----------------:|
> > | 0.1 | 80.00       | 47.75          |89.45           |
> > | 0.5       | 75.00        | 44.50   | 87.30          |
> > | 1        | 81.25 | 57.75 | **96.29** |
> >
> >
> > **Comment**: The experimental setting is unclear. For example, the network architecture is described by citing Nakkal & Salzmann (2021), but the authors should try to keep the description self-contained for reproducibility.
> >
> > **Answer**: The architecture of the generator is described below and has been added to Appendix E in the paper. For experiments on tabular data, we replace the convolution layers with fully connected layers.
> >
> > | Layer                                       | Number of Filters | Kernel Size |
> > |---------------------------------------------|-------------------|-------------|
> > | Convolution Layer                           | 32                | 9           |
> > | Convolution Layer                           | 64                | 3           |
> > | Convolution Layer                           | 128               | 3           |
> > | Residual Block                              | 128               | 3           |
> > | Residual Block                              | 128               | 3           |
> > | Residual Block                              | 128               | 3           |
> > | Upsampling Convolution Layer (Upsample = 2) | 64                | 3           |
> > | Upsampling Convolution Layer (Upsample = 2) | 32                | 3           |
> > | Convolution Layer                           | 32                | 9           |

---

### Official Review · Reviewer_JiWn · 2022-10-25

**Confidence:** 3
**Clarity, Quality, Novelty And Reproducibility:** The paper is original as far as I kno…
**Correctness:** 4
**Technical Novelty And Significance:** 4
**Empirical Novelty And Significance:** 3
**Recommendation:** 8

**Strength And Weaknesses:**

Strength:
1. The paper proposes a new paradigm other than contrastive learning for self-supervised learning.
2. The proposed algorithm is theoretically justified.
3. The paper is written and easy to follow in general.

Weakness:
1. The algorithm still requires domain-dependent hyperparameters (e.g., epsilon and L_p norm). I wonder if there is a way to automate the choices from the data which makes the whole algorithm agnostic.
2. The attack model is limited to additive attacks. However, some natural augmentation may require a very epsilon to realize using an additive model. For example, suppose a model would like to learn a translation-invariant representation (a shift in image/audio leads to the same semantic). In that case, a proper augmentation may be shifting or cropping if it is domain-specific, which is hard to realize by an additive model.
3. The theory seems a bit useless to me. The error bound depends on some factors that are not measurable in practice (e.g., alpha and lambda). I wonder if it is possible to obtain some numerical bound that measures whether the learned model succeeds or not (before evaluating it on test dataset).

**Summary Of The Paper:**

The paper proposes a domain-agnostic method to learn augmentations in self-supervised learning.

**Summary Of The Review:**

The paper proposed a simple and novel paradigm for self-supervised learning. Unlike contrastive learning, the proposed method can learn augmentation without domain knowledge. However, the paradigm still suffers from some limitations, as detailed in Weaknesses.

---

> ### Author Response · Authors · 2022-11-18
> **Response to Reviewer JiWn**
>
> We thank the reviewer for the insightful comments. Below is our point-to-point response.
>
> **Comment**: The algorithm still requires domain-dependent hyperparameters (e.g., epsilon and L_p norm). I wonder if there is a way to automate the choices from the data which makes the whole algorithm agnostic.
>
> **Answer**: We did sensitivity analyses w.r.t. hyperparameters of the model to demonstrate that the performance is robust to the hyperparameter choice. The sensitivity analyses can be found in Appendix C.2 in the paper. In addition, the number of tunable hyperparameters is small. As a result, we can easily tune the hyperparameters, possibly using hyperparameter tuning tools like Keras Tuner and Ray Tune.
>
> **Comment**: The attack model is limited to additive attacks. However, some natural augmentation may require a very epsilon to realize using an additive model. For example, suppose a model would like to learn a translation-invariant representation (a shift in image/audio leads to the same semantic). In that case, a proper augmentation may be shifting or cropping if it is domain-specific, which is hard to realize by an additive model.
>
> **Answer**:Currently, our model only uses additive noise, which may not realize some augmentations. Fortunately, our experiments show that additive noise achieves comparable performance to some natural augmentations on some image and audio datasets, with small epsilon. In future work, we plan to use conditional generative models to generate augmented views directly. The generated augmentations can realize more natural augmentations than additive noise.
>
> **Comment**: The theory seems a bit useless to me. The error bound depends on some factors that are not measurable in practice (e.g., alpha and lambda). I wonder if it is possible to obtain some numerical bound that measures whether the learned model succeeds or not (before evaluating it on test dataset).
>
> **Answer**: A relevant field of research to the mathematical analysis of machine learning model performance is Probably Approximately Correct (PAC) learning (Leslie, 1984). Our bound is derived by following the PAC learning framework. Although PAC learning is widely used in analyzing the theoretical performance of deep learning models (Saunshi et al., 2019, Tosh et al., 2021), it does not directly give a numerical bound. Our contribution is to derive a bound without making strong assumptions, yet providing insights.
>
> Valiant, Leslie G. "A theory of the learnable." *Communications of the ACM* 27.11 (1984): 1134-1142.
>
> Saunshi, Nikunj, et al. "A theoretical analysis of contrastive unsupervised representation learning." *International Conference on Machine Learning*. PMLR, 2019.
>
> Tosh, Christopher, Akshay Krishnamurthy, and Daniel Hsu. "Contrastive learning, multi-view redundancy, and linear models." *Algorithmic Learning Theory*. PMLR, 2021.

---

### Official Review · Reviewer_3qmD · 2022-11-04

**Confidence:** 3
**Correctness:** 3
**Technical Novelty And Significance:** 2
**Empirical Novelty And Significance:** 1
**Recommendation:** 5

**Clarity, Quality, Novelty And Reproducibility:**

- For the clarity, the paper is generally well-written with clear structures. It is quite easy to read.

- For the originality, it is a bit incremental in the sense that it simply replaces the widely used correlation / similarity measure (say cross-entropy) with another one. The introduction of adversarial training seems new, but its effectiveness needs more ablation study to justify.



**Strength And Weaknesses:**

For the strength:

- To the best of my knowledge, the idea seems to be new (although it sitll falls into the large category of non-contrastive learning, e.g., BYOL). The introduction of adversarial training seems interesting as well.

- The formulation is interesting, especially considering the similarity / correlation measure is a variational formulation which involves a constrained maximization as an inner optimization.

- The experiments show some improvement on a number of tasks including tabular data, vision data and audio data.


For the weakness:

- The novelty is incremental given that both the introduction of multi-dimensional Hirschfeld-Gebelein-Rényi (HGR) maximal correlation and adversarial traininng can not be well justified (both theoretically and empirically). It seems to be the idea is directly connected to Barlo Twins [https://arxiv.org/pdf/2103.03230.pdf] since the final objective is essentially similar. I think a proper comparison to Barlo Twins is necessary.

- The theoreical part does not inform why the proposed method is better than the others (say cross-entropy).

- Althoguh the paper presents a number of tasks, they are a bit toy-ish. It could much convincing if the authors can conduct some experiments on larger datasets (say ImageNet).


**Summary Of The Paper:**

The paper introduces a new self-supervised learning objective that uses a notation of multi-dimensional Hirschfeld-Gebelein-Rényi (HGR) maximal correlation as the similarity measure for different views of the same sample. The paper gives some theoretical insights and experimental study on some relatively small datasets.

**Summary Of The Review:**

I generally enjoy reading the paper and find the idea an interesting one. In terms of technical novelty, it is a bit incremental since the introduction of the new correlation measure is not well justified, as well as the adversarial training. In terms of empirical significance, I find it hard to evaluate based on the current toy-ish experiments.

My overall concerns are listed as follows:

- Since the final objective is a lagrangian relaxation, whether the final objective can well approximate the HGR correlation is unclear. I think the natural to evaluate this is to compute the individual loss terms in the objective and see how each term is optimized. For example, if the network output indeed tends to be element-wise independent, then the constraint for HGR correlation can be satisfied.

- For the novelty, as I mentioned above, it is a bit incremental. Just to add more details, I think the underlying motivation of the paper is very similar to Barlo Twins, but differently, the paper introduces a stronger regularization for the encoder network, ie, the network output is element-wise independent, which is essentially requiring the network to conduct something conceptually similar to nonlinear ICA. This is already a difficult task, and I don't understand why this will be beneficial for general data.

- For the experiments, I highly suggest the authors to conduct experiments on ImageNet and compare the results to Barlo Twins. Ideally, you can directly follow the settings of Barlo Twons, which can serve as a fair comparison.

Post-rebuttal:

Based on the comparison to Barlo Twins, the gain can be barely observed. I can see there are differences to Barlo Twins, but the authors do not really justify why these differences are generally beneficial to generalizability, and most importantly, the empirical evidence does not support the authors' argument. Given the above reason, I am still leaning towards rejection for the time being.

---

> ### Author Response · Authors · 2022-11-18
> **Response to Reviewer 3qmD**
>
> We thank the reviewer for the insightful comments. Below is our point-to-point response.
>
> **Comment**: The novelty is incremental given that both the introduction of multi-dimensional Hirschfeld-Gebelein-Rényi (HGR) maximal correlation and adversarial traininng can not be well justified (both theoretically and empirically). It seems to be the idea is directly connected to Barlo Twins [https://arxiv.org/pdf/2103.03230.pdf] since the final objective is essentially similar. I think a proper comparison to Barlo Twins is necessary.
> Ideally, you can directly follow the settings of Barlo Twins, which can serve as a fair comparison.
>
>
> **Answer**: The derivation of our method is different from Barlow Twins: our method is a traditional CCA/PLS method while Barlow Twins has a surrogate loss from information bottleneck.
>
> Specifically, Barlo Twins assumes that the representations follow a Gaussian Distribution so that the entropy of the representations can be computed analytically. In addition, they replace the determinant of covariance with the correlation matrix for empirical performance reasons.
>
> Our method differs from Barlow Twins also in some details. As shown in Figure 1, the two encoders are updated alternately, one by SGD and the other one by momentum. We get better performance due to the implicit ensemble and knowledge distillation effect due to momentum update.
>
> We do the comparison with Barlow Twins and BYOL. We use the image augmentations described in SimCLR. All models are trained for 200 epochs. Our method achieves comparable performance to BYOL and Barlow Twins. Results have been added to Appendix C in the paper.
>
>
> |                 | CIFAR-10 | CIFAR-100 | STL-10 | Tiny-ImageNet |
> |-----------------|----------|-----------|--------|---------------|
> | SimCLR          | 86.47    | 54.86     | 85.49  | 43.27 |
> | Barlow Twins    | 89.02    | **62.84**    | 85.43 | **45.33**       |
> | BYOL       | 88.54   | 61.76     | 85.59  | 42.75        |
> | Our            | **89.63**    | 62.55    | **86.41**  | 44.76 |
>
>
> **Comment**: The theoretical part does not inform why the proposed method is better than the others (say cross-entropy).
>
> **Answer**: Our theoretical analysis is not to show that the proposed method is better than other methods.  Our theorem is to guarantee the performance of the proposed method.
>
> Compared to theoretical analysis of other methods, our analysis requires fewer assumptions about the model and aligns better with practice. While the fundamental motivation behind contrastive learning is maximizing the mutual information between two augmented views (Oord et al., 2018), it is inconsistent with the actual behavior in practice, e.g., optimizing a tighter bound on mutual information can lead to worse representations (Tschannen et al., 2019). In the meantime, analysis based on the assumption of latent classes provides nice theoretical insights (Saunshi et al., 2019), but the theoretical result that representation quality suffers with more negative samples is inconsistent with empirical results in SimCLR and other popular contrastive learning methods. Wang et al, (2020) interpret contrastive loss as alignment and uniformity on the hypersphere, but they do not provide any theoretical guarantees on the downstream performance of the learned representation.
>
> We develop a self-supervised learning method based on well-known statistical learning methods such as CCA and PLS. Our goal is not to outperform SOTA but to give some insights about why training models with unlabeled data using certain loss functions leads to useful representations in the downstream task. We believe such insights are very important for the development of unsupervised representation learning in the long term.
>
> Oord, Aaron van den, Yazhe Li, and Oriol Vinyals. "Representation learning with contrastive predictive coding." arXiv preprint arXiv:1807.03748 (2018).
>
> Tschannen, Michael, et al. "On Mutual Information Maximization for Representation Learning." *International Conference on Learning Representations.* 2020.
>
> Saunshi, Nikunj, et al. "A theoretical analysis of contrastive unsupervised representation learning." *International Conference on Machine Learning*. PMLR, 2019.
>
> Wang, Tongzhou, and Phillip Isola. "Understanding contrastive representation learning through alignment and uniformity on the hypersphere." *International Conference on Machine Learning*. PMLR, 2020.

---

> > ### Author Response · Authors · 2022-11-18
> > **Response to Reviewer 3qmD (Continued)**
> >
> >
> > **Comment**: Althoguh the paper presents a number of tasks, they are a bit toy-ish. It could much convincing if the authors can conduct some experiments on larger datasets (say ImageNet).
> >
> > **Answer**: Unfortunately, we are not able to complete the experiment on ImageNet due to limited computation power and time constraints. Instead, we test our method on Tiny-ImageNet, which is a subset of ImageNet with 200 classes and still more challenging than CIFAR and STL. The performance of our method is slightly worse than SimCLR, but better than ViewMaker. APLR outperforms DACL by a large margin. Results on Tiny-ImageNet can be found in Table 2 in the paper.
> >
> >
> >
> > **Comment**: Since the final objective is a lagrangian relaxation, whether the final objective can well approximate the HGR correlation is unclear. I think the natural to evaluate this is to compute the individual loss terms in the objective and see how each term is optimized. For example, if the network output indeed tends to be element-wise independent, then the constraint for HGR correlation can be satisfied.
> >
> > **Answer**:
> >
> > To understand how well the final objective approximates the HGR correlation, we plot the two loss terms over training. Please find the figure in Appendix F. We find that the orthogonal loss approaches zero during training.

---

### Author Response · Authors · 2022-11-18
**Response to all reviewers**

We appreciate all reviewers for the comments to improve the quality of the paper.

The revised manuscript has been uploaded. We also made point-to-point responses to each reviewer. We are happy to make further responses if necessary.

---

### Comment · Area_Chair_pYdz · 2022-11-26
**Following up on authors’ response and discussion**

Dear Reviewers,

Thank you very much again for performing this extremely valuable service to the ICLR community.

Please check the authors’ response and leave comments if you have not done it.

Best,

AC

---

### Decision · Program_Chairs · 2023-01-20

**Decision:**

Reject

**Justification For Why Not Higher Score:**

While the motivation of self-supervised learning inspired by HGR maximal correlation is new and supported by theoretical guarantees, the contribution seems only incremental to the existing self-supervised learning literature. The Lagrangian-driven final objective resembles existing non-contrastive objectives, e.g., Barlow Twins [1] and Spectral Contrastive Loss [2]. In particular, one can easily check that the proposed objective is simply the Generalized Spectral Contrastive Loss [3] with EMA on the alignment term. Furthermore, the theoretical analysis of the guarantees for downstream generalization is quite similar to that of [2] (e.g., Assumption 4.1 & 4.2 to Assumption 3.6 & 3.7 in [2]). However, the paper does not contain such a comparison between other contrastive objectives, especially missing the comparison with spectral contrastive loss, which seems the most relevant reference. Also, as Reviewer 3qmD mentioned, the proposed objective does not show significant gain over other SSL methods, e.g., Barlow Twins, and the theoretical guarantee does not guarantee better generalizability.

The paper has demonstrated the empirical advantage of the proposed method in various domain-agnostic self-supervised learning tasks, however, the empirical gain is not significant or the proposed baselines are not enough to claim the superiority of the proposed method.
For example, as Reviewer 3qmD mentioned, the paper considers only small-scale dataset while most of the self-supervised literature consider large-scale ImageNet dataset for evaluation.

Moreover, there are some missing baselines in domain-agnostic SSL. For example, I-MIX [4] is a mixup-based contrastive learning method that has shown its effectiveness in domain-agnostic SSL on vision, tabular, and speech datasets. Furthermore, MAE [5] is also a popular self-supervised learning method that can be used in a domain-agnostic way as illustrated in [6, 7].

Overall, the paper proposed an interesting direction for self-supervised learning objectives and demonstrated its effectiveness on domain-agnostic self-supervised learning tasks, but AC thinks the limited technical contribution and empirical findings are not significant enough for publication.

[1] Zbontar, Jure, et al. "Barlow twins: Self-supervised learning via redundancy reduction." International Conference on Machine Learning. PMLR, 2021.

[2] HaoChen, Jeff Z., et al. "Provable guarantees for self-supervised deep learning with spectral contrastive loss." Advances in Neural Information Processing Systems 34 (2021): 5000-5011.

[3] HaoChen, Jeff Z., et al. "Beyond separability: Analyzing the linear transferability of contrastive representations to related subpopulations." arXiv preprint arXiv:2204.02683 (2022).

[4] Lee, Kibok, et al. "i-mix: A domain-agnostic strategy for contrastive representation learning." arXiv preprint arXiv:2010.08887 (2020)

[5] He, Kaiming, et al. "Masked autoencoders are scalable vision learners." Proceedings of the IEEE/CVF Conference on Computer Vision and Pattern Recognition. 2022.

[6] Tamkin, Alex, et al. "Dabs: A domain-agnostic benchmark for self-supervised learning." arXiv preprint arXiv:2111.12062 (2021).

[7] Tamkin, Alex, et al. "DABS 2.0: Improved Datasets and Algorithms for Universal Self-Supervision." Thirty-sixth Conference on Neural Information Processing Systems Datasets and Benchmarks Track. 2022.



**Justification For Why Not Lower Score:**

N/A

**Metareview: Summary, Strengths And Weaknesses:**

This paper proposes a new self-supervised learning objective and its application to domain-agnostic SSL by using adversarial view generation. The proposed method is grounded by HGR maximal correlation, which interpolates between CCA and PLS. Then, they propose a novel SSL objective that has a theoretical guarantee for downstream generalization. Together with adversarial view generation by training auxiliary generator, they empirically validate the effectiveness of the proposed method on various domain-agnostic SSL benchmarks on vision, tabular data, and speech datasets.

Strength:
- This paper proposes a novel non-contrastive self-supervised learning framework derived from maximal HGR correlation appended with theoretical support. (All reviewers agreed)
- Marginal improvement in domain-agnostic self-supervised learning baselines on tabular data, vision, and audio datasets. (All reviewers agreed)

Weakness:
- Missing the relationship/comparison between the proposed SSL objective and existing SSL objectives. See the below comment. (Reviewer 3qmD, Hv4e, RgE2)
- The theoretical analysis seems vacuous (Reviewer 3qmD, JiWn)
- Lack of large-scale experiments, e.g., ImageNet (Reviewer 3qmD)